# (1D) Ordered Tokens Enable Efficient Test-Time Search

**Zhitong Gao**[1]  **Parham Rezaei**[1]  **Ali Cy**[1]  **Mingqiao Ye**[1]  **Nataša Jovanović**[1]  **Jesse Allardice**[2]  **Afshin Dehghan**[2]  **Amir Zamir**[1]  **Roman Bachmann**[*1 2]  **Oğuzhan Fatih Kar**[*1 2]

https://soto.epfl.ch

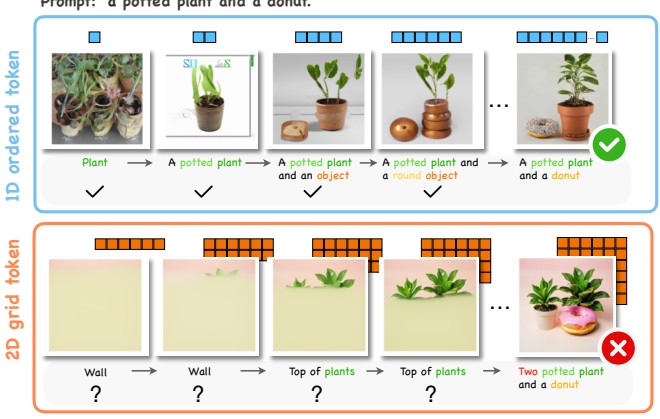

(a) 1D ordered tokens provide an **interpretable readout** with a **coarse-to-fine structure** amenable to test-time search.

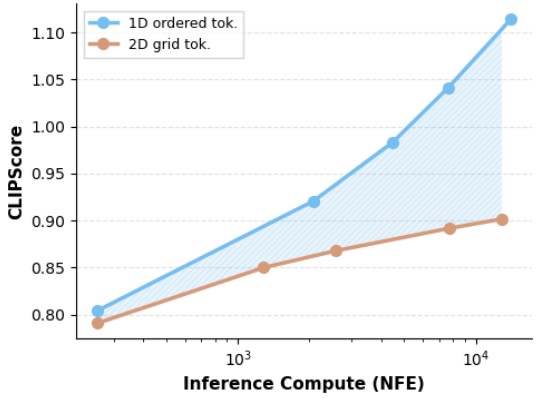

(b) Models trained with 1D ordered tokens **benefit more from increased test-time compute**.

*Figure 1.* (a) **Intermediate readouts.** 1D ordered tokens provide a coarse-to-fine structure with interpretable readouts amenable to test-time search. For the prompt *'a potted plant and a donut"*, tokens progressively capture concepts from high- to low-level, e.g., *'plant"* → *'potted plant"* → *'a potted plant and an object"*. This structure allows verifiers to effectively guide generation. In contrast, 2D grid tokens generated in raster-scan order are harder to verify and search over. (b) **Scaling behavior.** 1D ordered tokens (in this plot, FlexTok (Bachmann et al., 2025)) exhibit better test-time scaling than 2D grid tokens (from a controlled baseline) when using the best search algorithm for each (beam search for 1D and best-of-N for 2D). See Figure 6 for complete results.

## Abstract

Tokenization is a key component of autoregressive (AR) generative models, converting raw data into more manageable units for modeling. Commonly, tokens describe local information, such as regions of pixels in images or word pieces in text, and AR generation predicts these tokens in a fixed order. A worthwhile question is whether token structures affect the ability to steer the generation through test-time search, where multiple candidate generations are explored and evaluated by a verifier. Using image generation as our testbed, we hypothesize that recent 1D ordered tokenizers with

coarse-to-fine structure can be more amenable to search than classical 2D grid structures. This is rooted in the fact that the intermediate states in coarse-to-fine sequences carry semantic meaning that verifiers can reliably evaluate, enabling effective steering during generation. Through controlled experiments, we find that AR models trained on coarse-to-fine ordered tokens exhibit improved test-time scaling behavior compared to grid-based counterparts. Moreover, we demonstrate that, thanks to the ordered structure, pure test-time search over token sequences (i.e., without training an AR model) can perform training-free text-to-image generation when guided by an image-text verifier. Beyond this, we systematically study how classical search algorithms (best-of-$N$, beam search, lookahead search) interact with different token structures, as well as the role of different verifiers and AR priors.

*Equal technical advising. [1]Swiss Federal Institute of Technology Lausanne (EPFL) [2]Apple. Correspondence to: Zhitong Gao <zhitong.gao@epfl.ch>, Roman Bachmann <r_bachmann@apple.com>, Oğuzhan Fatih Kar <o_kar@apple.com>.

*Proceedings of the 43rd International Conference on Machine Learning*, Seoul, South Korea. PMLR 306, 2026. Copyright 2026 by the author(s).

# 1. Introduction

Autoregressive generative models rely on tokenization to convert raw data into more compact modeling units. The most common approaches across modalities encode information *locally*, such as images into spatial grids (van den Oord et al., 2017; Esser et al., 2020; Sun et al., 2024) where local clusters of tokens correspond to local regions of pixels, or text into subwords (Song et al., 2021; Kudo & Richardson, 2018). Autoregressive generation then predicts these tokens sequentially, following spatial orderings like raster-scan for images or left-to-right for text. An important question is how these structural choices affect the model's ability to perform test-time search, where generation explores multiple candidates guided by a verifier, a technique that has proven valuable for improving generation quality and control in language modeling (Lightman et al., 2023; Yao et al., 2023; Wei et al., 2022) and diffusion models (Ma et al., 2025a; Singhal et al., 2025; Zhang et al., 2025).

One can speculate that token structure matters for search. To investigate this, we study recent 1D ordered tokenizers (Bachmann et al., 2025; Wen et al., 2025b), which compress images into sequences with an ordering that reflects a coarse-to-fine or semantic-to-detailed structure. Compared to grid-based representations, these tokenizers produce semantically meaningful intermediate readouts and support detokenization from a variable number of tokens.

The key observation motivating this work is that in coarse-to-fine orderings, intermediate sequences of generated tokens carry *global* semantic meaning that can be reliably evaluated by verifiers, enabling pruning and refinement via search. In contrast, spatially-ordered tokenizers produce tokens that correspond only to fixed spatial regions (e.g., the upper-left corner of an image), which provide limited semantic signal about the full output. Consider the example in Figure 1. Using 1D ordered tokenizers like FlexTok (Bachmann et al., 2025), the choice of the first token significantly narrows down the space of possible generations towards ones that show plant-like concepts, and the ordered tokenizer's intermediate readouts show semantic content throughout generation. In contrast, the 2D grid tokenizer's first tokens only reveal the upper-left spatial region featuring a wall.

In this paper, we specifically focus on autoregressive image generation as our testbed. While image tokenizers have primarily been evaluated through reconstruction and generation fidelity (Sun et al., 2024; Yu et al., 2024), we adopt a complementary perspective: their *test-time scaling (TTS) behavior*, i.e., their ability to improve generation quality and alignment through search-based inference.

We demonstrate that an autoregressive model trained on 1D ordered tokens exhibits stronger test-time scaling behavior than a comparable AR model trained on 2D grid

tokens. To stress-test the role of token structure, we show that pure test-time search over ordered token sequences can enable training-free (i.e., without training an autoregressive model) text-to-image generation when guided by an image-text similarity verifier. We further show that an AR model trained solely with text-conditioning can perform image-controlled generation in a training-free manner when paired with an image-image similarity verifier. Finally, to understand the broader design space, we provide a comprehensive analysis of how different search strategies (beam search, best-of-$N$ sampling, lookahead search) perform across token structures, and investigate how different verifiers and autoregressive priors influence search-guided generation.

Code, additional interactive visualizations, and model weights are available at `https://soto.epfl.ch`.

**Conflict of Interest Disclosure.** One of the models this paper evaluates is FlexTok (Bachmann et al., 2025), which was developed by members of the authorship team affiliated with Apple. The FlexTok tokenizer is publicly available.

# 2. Background

In this section, we review AR generation, test-time search, and 1D ordered tokenizers to provide the necessary context for our work, followed by our core motivation. A comprehensive discussion of related literature is deferred to Sec. 6.

**Autoregressive generation** AR generation is a standard paradigm for generative modeling across text, image, and multimodal domains (Touvron et al., 2023; Ramesh et al., 2021; Wu et al., 2024a). It typically *tokenizes* data into discrete tokens $\mathbf{x} = (x_1, \ldots, x_T)$, where $x_t \in \mathcal{V}$, and models the data distribution via next-token prediction:

$$p_\theta(\mathbf{x} \mid c) = \prod_{t=1}^{T} p_\theta(x_t \mid x_{<t}, c), \tag{1}$$

where $c$ is an optional conditioning context.

**Test-time search** While standard inference performs greedy decoding or sampling from the learned distribution, an alternative is to treat the learned AR model as a prior and conduct verifier-guided search at inference time: selecting the sequence $\hat{\mathbf{x}}$ that maximizes a *verifier* $g(\mathbf{x}, c) := S(\text{Dec}(\mathbf{x}), c)$, where $\text{Dec}(\cdot)$ maps tokens to pixel space and $S$ is a similarity metric:

$$\hat{\mathbf{x}} = \arg\max_{\mathbf{x}} \ g(\mathbf{x}, c) \quad \text{s.t.} \quad x_t \in \mathcal{K}(p_\theta(\cdot \mid x_{<t}, c)). \tag{2}$$

Here, $\mathcal{K}$ restricts the search space to likely candidates (e.g., top-$k$). Algorithms like *Best-of-$N$*, *Beam Search*, and *Lookahead Search* (Snell et al., 2024) represent different strategies for approximating this constrained optimization.

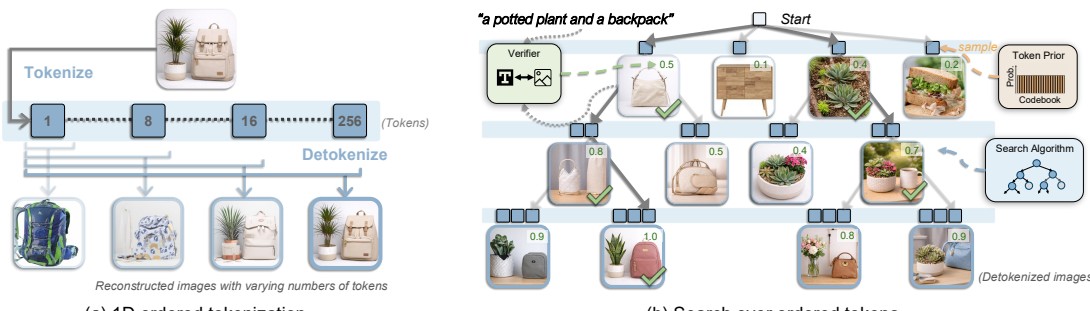

(a) 1D ordered tokenization        (b) Search over ordered tokens

*Figure 2.* **Ordered tokens induce a searchable latent structure.** (a) FlexTok encodes images into a sequence of 1D ordered tokens trained to support variable-length decoding, imposing a coarse-to-fine hierarchy. (b) Illustration of search over the token vocabulary without an autoregressive model: candidate tokens are sampled using a *token prior* (here, uniform over the codebook, i.e., no AR model is assumed) and evaluated using a *verifier* (e.g., CLIP); a *search algorithm* (here, beam search) then expands the most promising partial sequences. As more ordered tokens are included, intermediate reconstructions progressively refine global semantics and visual details. See Fig. 5 for a full description of the token prior, verifier, and search algorithm components used in our framework.

Test-time search enables more reliable generation through verification, and is a popular direction to achieve *test-time scaling* as it trades additional inference compute for generation quality (Brown et al., 2024; Ma et al., 2025b).

**1D ordered tokenizers.** AR generation for images has co-evolved with tokenizer design. The standard approach encodes images into fixed 2D grid tokens, which implicitly assumes information is distributed uniformly across space. 1D *ordered* tokenizers like FlexTok (Bachmann et al., 2025) and Semanticist (Wen et al., 2025a) instead encode images into flexible-length 1D sequences, typically trained with nested dropout (Rippel et al., 2014). This ensures that any prefix can be decoded into a valid image, with early tokens capturing global structure and later tokens refining details.

**Motivation** Existing work on test-time scaling has largely focused on designing better search algorithms (Chen et al., 2025c), stronger verifiers (Lightman et al., 2023), or studying scaling laws (Snell et al., 2024), but less attention has been paid to what characteristics a model should *possess* to benefit from test-time scaling in the first place. We argue that token structure is a key factor: it defines the search space and how intermediate states connect, and thus determines how verifiable each state is. Below, we first analyze why this token structure is more amenable to search (§3), and then introduce a systematic framework to study test-time scaling across different tokenizer designs (§4).

## 3. Coarse-to-Fine Ordered Token Structures are More Amenable to Search

We hypothesize that 1D ordered tokens induce a hierarchical, coarse-to-fine structure that makes the token space more amenable to search. In this section, we investigate the latent structure of 1D ordered tokenizers (specifically FlexTok (Bachmann et al., 2025)) to validate this suitability.

### 3.1. The First Token is a Global Semantic Cluster

We begin by examining the information density of the first token. In standard 2D VQGAN-based tokenization, the first token corresponds to a local pixel patch in the top-left corner and thus contains little semantic information about the entire image. In contrast, the first token in FlexTok is trained to reconstruct the *entire* image at a high compression ratio, encouraging it to capture global semantics.

To visualize this property, we select different first tokens from a vocabulary of 64K entries, and decode each multiple times using different random seeds. Figure 3 shows example reconstructions, where each token is decoded nine times. The resulting images form semantically coherent clusters (e.g., plants, bags, food, and furniture), indicating that individual first-token entries correspond to meaningful global semantic categories. As demonstrated by Bachmann et al. (2025), subsequent tokens model ever more detailed *"concepts"*, narrowing down the distribution over images defined by the token sequences.

### 3.2. Zero-Shot Generation via Pure Search

If the token space is semantically ordered, it should be possible to generate images by directly searching for tokens that maximize alignment with a text prompt, even in the absence of a generative model. We test this hypothesis by performing *beam search* directly over the token space. At each step $t$, we expand the top-$k$ partial token sequences, detokenize them into images, and rank the candidates based on their CLIP (Radford et al., 2021) or ImageReward (Xu et al., 2023) similarity to the text prompt. An illustration of this procedure is shown in Fig. 2.

Figure 4 visualizes the images obtained during this search process. We observe that this approach produces coherent and semantically aligned images, with progressively finer details emerging as search explores additional tokens.

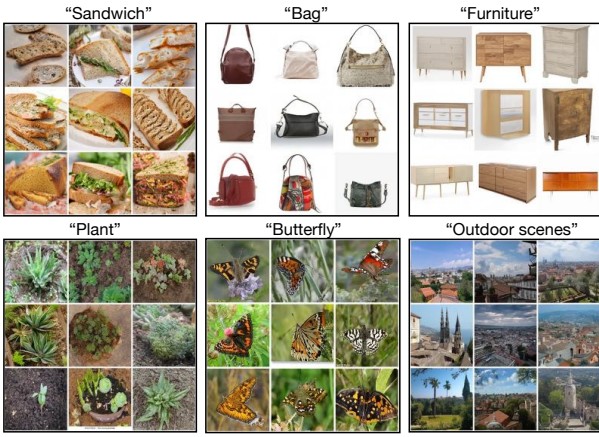

**Figure 3. Visualization of images decoded from the first-token vocabulary in FlexTok.** Each first-token entry is decoded using nine random seeds, producing nine images per token. These decoded images form semantically coherent clusters (e.g., plants, bags, food, and furniture), indicating that tokens capture a global *distribution of concepts* that can be searched over.

Notably, this behavior does not arise with 2D grid tokenizations or 1D *unordered* tokenizers (Yu et al., 2024); without a coarse-to-fine structure, earlier tokens provide little information about later ones, making a full search over all combinations computationally infeasible.

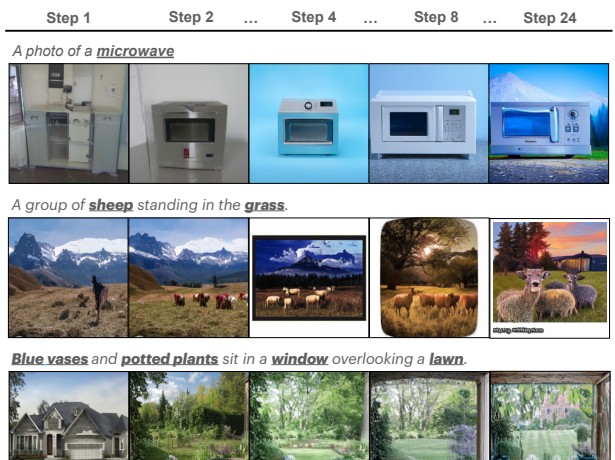

**Figure 4. Direct search over 1D ordered tokens enables training-free text-to-image generation.** We search over FlexTok (Bachmann et al., 2025) using a 5-beam strategy and ImageReward as the verifier. We show the best image obtained at each step.

### 3.3. A Theoretical Perspective

Searching over token sequences to maximize a verifier score is analogous to nearest-neighbor search in structured data, where search efficiency depends critically on how the space is organized. Just as kd-trees (Bentley, 1975) or PCA-based partitions (McNames, 2002) enable efficient retrieval by structuring data along the most informative directions,

search over tokens benefits when earlier tokens capture the global semantics that verifiers rely on. As formalized in Appendix B, 1D ordered tokenizers use nested dropout (Rippel et al., 2014) to explicitly minimize the reconstruction error of intermediate decoded images. Under a Lipschitz assumption on the verifier, this minimization directly bounds the heuristic error (Pearl, 1983) at each step, yielding a tighter theoretical bound on the overall search gap.

Both empirical and theoretical results suggest that the coarse-to-fine structure of 1D ordered tokens makes them more amenable to search, as intermediate representations can be effectively evaluated by verifiers that capture global semantic properties. In Sec. 4, we build on this to study search over tokens with autoregressive models.

## 4. Search-over-Tokens (SoTo) Framework

In this section, we investigate test-time search combined with autoregressive modeling. To systematically evaluate scaling ability, we employ three standard *search algorithms* (best-of-$N$, beam, and lookahead search) and assess performance across nine different *verifiers*. Finally, we analyze the impact of the autoregressive prior by varying the guidance level from strong (text-conditional) to weak (unconditional) to none, testing whether an effective tokenizer enables search with minimal priors. We call the approach in this study "**S**earch-**o**ver-**To**kens" (SoTo). Below, we discuss the three components in more detail.

**Search algorithm.** We consider three popular search algorithms and combine them with AR image generation models. (1) *Best-of-$N$ sampling* generates $N$ independent sequences from the AR model and selects the one with the highest verifier score; it is simple and parallelizable but does not leverage structure or partial tokens in the search trajectory, and thus can sometimes be inefficient. It is usually used as a baseline for test-time scaling. (2) *Beam search* instead maintains a set of $k$ partial hypotheses (beams), expanding each using the model's top-$M$ next tokens (candidates) and retaining the best-scoring continuations. As it operates on partial token sequences, the informativeness of these intermediate states is critical. For 2D grid tokenizations, to enable detokenization from partial token sequences, we pad the remaining tokens, as illustrated in Figure 1. (3) Besides these two, we also explore *Lookahead search*, which is based on beam search but further improves reliability by rolling out partial sequences into more complete images before scoring them, providing verifiers with more meaningful inputs. This approach is especially useful for 2D tokens, but it also makes each decision step significantly more computationally expensive.

In practice, we perform beam search and lookahead search at intermediate token positions and directly generate the re-

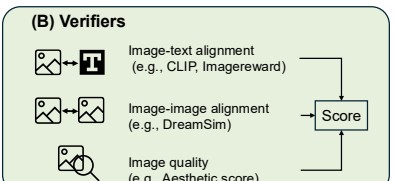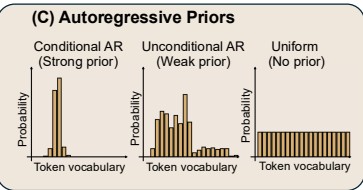

*Figure 5.* **Overview of the Search-over-Tokens (`SoTo`) evaluation framework.** The framework studies test-time scaling behavior of image tokenizers when combined with autoregressive generation and search. (A) *Search algorithms*: different strategies for exploring the token space during generation, including Best-of-$N$ sampling, Beam Search, and Lookahead Search. (B) *Verifiers*: scoring functions that guide search by evaluating partial or complete decoded images, covering image–text alignment, image–image alignment, and image quality objectives. (C) *Autoregressive prior*: next-token probability models that constrain the search space, ranging from text-conditional AR models to unconditional AR models and uniform (prior-free) baselines.

maining tokens with the AR model. We vary the number of search tokens to control the compute budget. An illustration of these three search algorithms is provided in Figure 5 (A), and additional details are provided in Appendix C.1.

**Verifier.** Verifiers serve as the search objective, guiding generation by scoring partial or complete token sequences. We consider three major categories: *(1) image–text alignment*, which assesses correspondence between an image and a prompt using models such as CLIP (Hessel et al., 2021), ImageReward (Xu et al., 2023), CycleReward (Bahng et al., 2025), PickScore (Kirstain et al., 2023) , and HPSv2 (Wu et al., 2023). We also explore using the self-likelihood of tokens in the guided AR model, and design a rule-based verifier using a segmentation model (e.g., Grounded SAM (Ren et al., 2024)). *(2) image–image alignment*, which measures similarity to a reference image using models such as Dream-Sim (Fu et al., 2023); and *(3) image quality* verifiers, such as aesthetic predictors (Schuhmann et al., 2022), which evaluate fidelity or aesthetics independent of text.

These verifiers capture different aspects of image quality and are often complementary, for example, rule-based verifiers provide precise checks on object presence or position, while CLIP-like models better capture global semantics such as color or style. Therefore, we also explore *ensemble verifiers* by combining multiple signals through rank-based aggregation for more robust guidance. An illustration of the verifier categories we explored is shown in Fig. 5 (B). Please see the App. C.2 for implementation details of different verifiers.

**Autoregressive prior.** Lastly, we consider the role of the AR model, which provides prior probabilities for the next token and helps prune the search space to its most likely regions. To study its role, we compare performance using the original text-conditional AR model, an unconditional AR model (the same AR model but without text conditioning), and a no-AR model (uniform prior). Intuitively, these configurations represent a spectrum of guidance: the text-*conditional* model provides the strongest prior, followed by the *unconditional* model, while the *uniform* prior is the weakest. An illustration of these priors is shown in Fig. 5

(C) and additional details are provided in App. C.3.

## 5. Experiments

In this section, we study *1D ordered tokenization from a test-time search perspective*. We first describe the setup, then present results on test-time scaling with different search algorithms, verifier-guided zero-shot control, generation under different priors, and a comprehensive verifier analysis.

### 5.1. Experimental Setting

**Models.** Our primary 1D ordered tokenizer is FlexTok d18–d28, paired with a default 3.4B-parameter AR model. For scaling analysis, we additionally evaluate smaller AR variants (212M, 530M, and 1.4B parameters). For controlled comparisons, we employ a 2D grid tokenization baseline (Bachmann et al., 2025) that exactly matches the data, architecture, and training compute of our 3.4B FlexTok setup. We also evaluate Janus-1.3B (Wu et al., 2024a), a competitive 2D grid-based AR model. Finally, to demonstrate generalizability, we evaluate other ordered generation models (Semanticist (Wen et al., 2025a) and Infinity (Han et al., 2025)) in Appendix E.1.

**Datasets.** We evaluate text-to-image generation on the COCO Karpathy validation set (Lin et al., 2014) and GenEval (Ghosh et al., 2023). For zero-shot multimodal control, we use DreamBench++ (Peng et al., 2024), which includes reference images in addition to text prompts.

**Search Algorithms.** We evaluate best-of-$N$ sampling, beam search, and lookahead search. Unless otherwise specified, we vary $N$ from 1 to 50 for best-of-$N$. For beam and lookahead search, we use a beam width of 5 with 10 candidates per step and vary the number of searched tokens.

**Inference Compute.** We report performance as a function of inference compute measured by the *Number of Function Evaluations* (NFE), counting each token generation or verification as one evaluation (cf. App. D.3 for details).

More implementation details are provided in App. C and D.

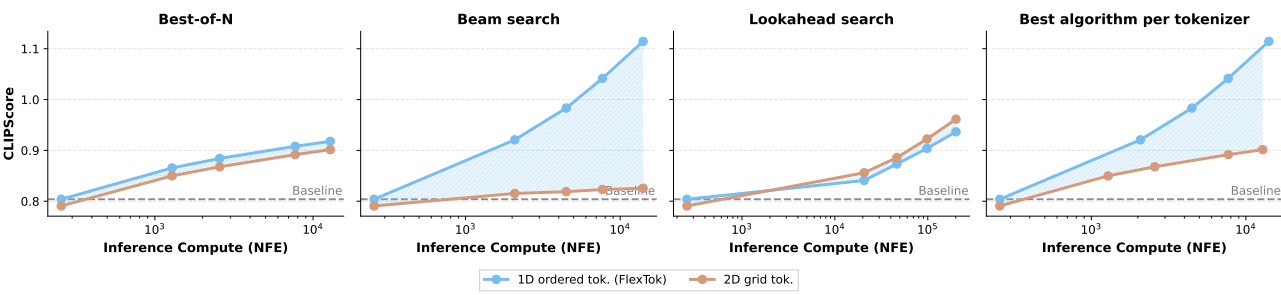

*Figure 6.* **Test-time scaling across token structures.** We compare inference-time search algorithms on two tokenizers: 1D ordered tokens (FlexTok) and a controlled 2D grid tokenizer. While best-of-$N$ and lookahead search exhibit similar scaling for both tokenizations, beam search yields substantially larger gains for 1D ordered tokens. *The rightmost panel compares each tokenizer under its best-performing search algorithm, showing that 1D ordered tokens benefit more from search.* Results are evaluated on the COCO Karpathy validation set (Lin et al., 2014). NFE denotes the number of function evaluations; the leftmost point corresponds to no search. The dashed line indicates the no-search baseline for FlexTok and is extended for reference.

## 5.2. 1D Ordered Tokens Enable Better TTS

**Controlled Experiments** To isolate the effect of token structure on test-time scaling, we compare AR models trained on 1D ordered tokens (FlexTok) and a controlled 2D grid token baseline. We evaluate the search strategies described above; see Figure 6. For best-of-$N$, we use $N \in \{1, 5, 10, 30, 50\}$. For beam search, we vary the number of searched tokens in $\{16, 64, 128, 256\}$, and for lookahead search, we perform full rollouts with searched tokens in $\{4, 8, 16, 32\}$. Due to the high cost of large search budgets, experiments are conducted on a 300-image subset of the COCO Karpathy validation set (Lin et al., 2014), which we find sufficient and stable (cf. App. E.3).

Under no search, the two models achieve comparable CLIP scores, confirming that they have similar base generation quality. As inference compute increases, both tokenizations exhibit similar scaling trends under best-of-$N$ sampling and lookahead search. In contrast, *beam search produces markedly different behavior across token structures*: while performance improves rapidly for 1D ordered tokens, it yields only marginal gains for 2D grid tokens.

This divergence indicates that the effectiveness of search depends critically on token structure: for 1D ordered tokens, partial prefixes already encode semantically meaningful global structure, making *beam search the most compute-efficient strategy*. For 2D grid tokens, intermediate states provide weak or misleading signals, making beam search ineffective. While lookahead search can partially recover performance by rolling out more complete images before scoring, it comes at substantially higher computational cost. Within a comparable compute budget, *best-of-$N$ sampling* therefore achieves the best performance for 2D grid tokens.

Finally, when comparing each tokenizer under its best-performing search algorithm, *1D ordered tokenization consistently achieves higher performance* across inference budgets. This demonstrates a clear representation-level advan-

tage: the gap between 1D and 2D tokenizations cannot be closed by search algorithm choice alone. Consistent trends are observed on GenEval (cf. App. E.2).

**Comparison with Janus.** We further compare FlexTok with Janus (Wu et al., 2024a), a competitive autoregressive model using a 2D grid tokenizer. The two models are closely matched in base performance, making this comparison well suited for examining differences in test-time scaling.

As shown in Fig. 7, although Janus achieves slightly higher performance without search, *FlexTok exhibits stronger test-time scaling under beam search*. Consistent with our controlled experiments, FlexTok leverages beam search more effectively, demonstrating the scaling advantage of 1D ordered tokenization. In contrast, beam search provides limited benefits for Janus, where best-of-$N$ sampling slightly outperforms it across inference budgets.

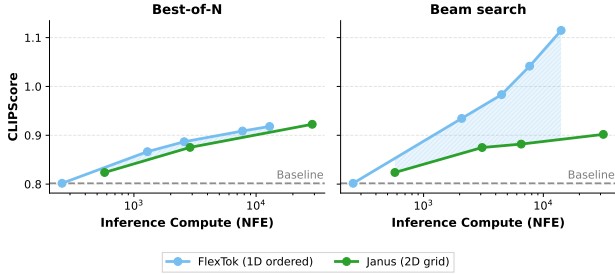

*Figure 7.* **Test-time scaling compared with Janus.** We compare FlexTok (1D ordered tokens) and Janus (Wu et al., 2024a) (2D grid tokens) under best-of-$N$ sampling and beam search. While Janus achieves slightly higher performance without search, FlexTok exhibits stronger scaling under beam search as inference compute increases. Results are evaluated on the COCO validation set.

**Other ordered generation paradigms.** To further verify that our findings generalize beyond FlexTok, we evaluate two additional ordered generation paradigms. First, we consider **Semanticist** (Wen et al., 2025a), a 1D ordered tokenizer that shares FlexTok's nested-dropout-based ordering

mechanism but differs in architecture and token space design, and compare it against LlamaGen (Sun et al., 2024), a comparable model using 2D grid tokens, on ImageNet-1K class-to-image generation. We observe the same trend as in our experiments: while all search methods improve both models, *beam search yields larger gains for the ordered 1D tokenizer*. Second, we evaluate **Infinity** (Han et al., 2025), a scale-wise autoregressive model related to VAR (Tian et al., 2024), on text-to-image generation on COCO. Infinity benefits more from search than Janus but less than FlexTok, suggesting that *ordering improves search effectiveness, while semantic coarse-to-fine ordering is particularly effective*. Detailed results are provided in App. E.1.

**Scaling across model sizes.** We examine the extent to which inference-time search can compensate for training-time compute. We evaluate FlexTok autoregressive models across a range of parameter sizes using best-of-$N$ sampling, which provides a consistent way to control inference compute and trace scaling behavior across model sizes.

We observe that *a 530M-parameter model with sufficient test-time compute can outperform a larger 3.4B-parameter model operating with limited inference compute* (Fig. 8). As inference compute increases, however, larger models exhibit stronger scaling behavior. Overall, performance traces a Pareto frontier with respect to inference FLOPs, where the optimal model size increases with the available compute budget and follows a power-law relationship.

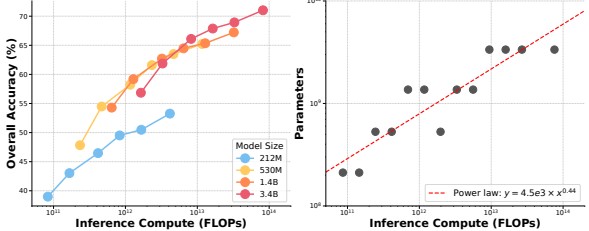

*Figure 8.* **Performance of search across different model sizes.** We study test-time scaling with different FlexTok AR sizes. **(Left):** We use GenEval with best-of-$N$ search and estimate the corresponding inference FLOPS. **(Right):** Extracting the model size with best performance within equally log spaced FLOPs buckets, we find alignment with a power law relationship. Fitting a power law of the form $y = a \times x^b$ for the optimal model size as a function of inference compute, we find $a = 4.5 \times 10^3$ and $b = 0.44$.

### 5.3. 1D Ordered Tokens with Zero-Shot Control

Beyond test-time scaling, we find that search over 1D ordered tokens enables *zero-shot control* using conditioning signals not seen during training. We study a setting where the model generates an image from a text prompt while preserving a visual concept from a reference image, despite being trained only on text–image pairs.

We perform beam search with FlexTok using an image–

image similarity verifier, DreamSim (Fu et al., 2023), and compare against Janus. To provide dense guidance, we apply search over the first 32 tokens for both models; for Janus, we use full lookahead rollouts at each step, which are necessary for concept preservation with 2D grid tokens, with all other parameters unchanged. We evaluate on Dream-Bench++ (Peng et al., 2024), a concept preservation benchmark with text prompts and reference images.

As shown in Fig. 9 and Table 1, search substantially improves concept preservation for FlexTok (+18.4 on DINO-I) while preserving prompt-following performance. Janus also benefits, but the gains are smaller (+5.9 on DINO-I), even with lookahead search, indicating that *1D ordered tokenization more effectively supports zero-shot image-guided control*. Additional qualitative results are shown in App. F.3.

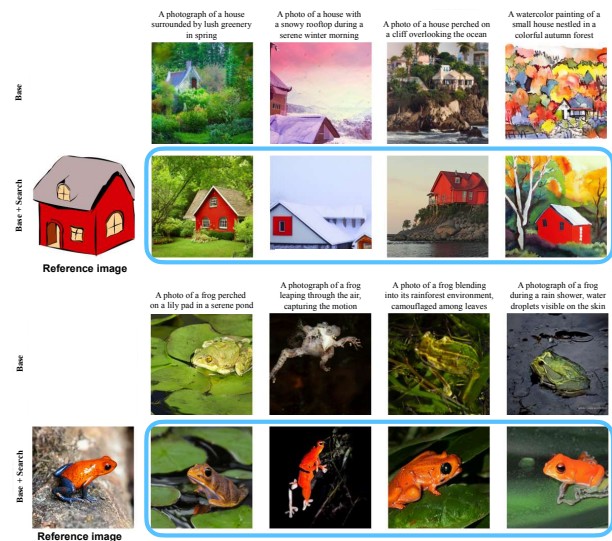

*Figure 9.* **Image generation with zero-shot concept preservation via search.** Search over 1D ordered tokens enables multimodal control without finetuning by incorporating an image similarity verifier (DreamSim (Fu et al., 2023)) at inference time. The top row shows direct autoregressive generation with FlexTok, while the bottom row shows generations guided by image-based verification.

*Table 1.* **DreamBench++ concept preservation and prompt-following results.** We follow the benchmark and report improvements in concept preservation (DINO-I, CLIP-I) and prompt following (CLIP-T) for FlexTok and Janus under test-time search. Both models benefit from search, but FlexTok achieves substantially larger gains with lower test-time compute.

| Method | DINO-I | CLIP-I | CLIP-T |
|---|---|---|---|
| FlexTok | 32.5 | 68.1 | 34.1 |
| FlexTok + Beam Search | **50.9** +18.4 | **76.5** +8.4 | 33.1 |
| Janus | 34.8 | 69.0 | 35.5 |
| Janus + Lookahead Search | **40.7** +5.9 | **71.9** +2.9 | 35.6 |

## 5.4. 1D Ordered Tokens Enable Generation by Search

As discussed in Sec. 3, direct beam search over FlexTok tokens can generate reasonable images. We quantitatively evaluate this *generation-by-search* setting and analyze the role of the autoregressive prior. Specifically, we consider three priors: (1) a *conditional* AR prior (standard text-conditioned FlexTok), (2) an *unconditional* AR prior, and (3) a *uniform* prior. For the unconditional and uniform priors, we uniformly sample 10% of the token space during search.

We evaluate on a subset of 180 GenEval prompts covering single- and two-object categories. As shown in Table 2, search with a uniform prior achieves 79% on single-object generation and 32% on two-object generation, demonstrating that generation without any AR prior is feasible. Incorporating an unconditional prior further improves performance, while the conditional prior achieves the best results overall. Fig. 10 and App. F.1 provide visual examples.

Similarly, experiments with another 1D ordered tokenizer (Semanticist (Wen et al., 2025a)) show that text-to-image generation remains feasible even with a weak AR prior (e.g., a class-conditional prior); see App. E.1.

*Table 2.* **Quantitative comparison of three different priors for search.** We compare the performance of a uniform prior, an unconditional AR prior, and a conditional AR prior using beam search on the GenEval subset with FlexTok. Results in Acc. (%).

| Prior | Search | Single Object | Two Object |
|---|---|---|---|
| Uniform | yes | 79 | 32 |
| Unconditional | yes | 85 | 33 |
| Conditional | yes | **100** | **81** |
| Conditional | no | 97 | 48 |

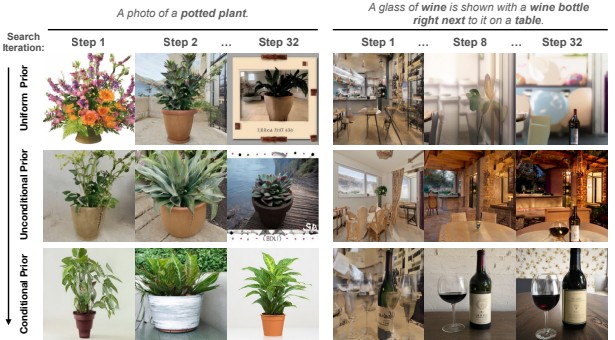

*Figure 10.* **Visual comparison of three priors for search.** Uniform and unconditional priors combined with verifier-guided test-time search can generate reasonable images, but they explore a much larger search space than the conditional prior. As a result, the conditional prior often reaches tokens closer to the final image at earlier steps (e.g., directly identifying a potted plant or bottle in the first token). We show two example prompts of different complexity: simple prompts can often be matched within two tokens, while more complex prompts require longer token sequences.

## 5.5. Analysis of Different Verifiers

Finally, we study test-time scaling with different verifiers. All experiments are conducted on GenEval with FlexTok and beam search, using 9 search steps to balance efficiency and performance. We evaluate eight verifiers: likelihood, CLIPScore (Radford et al., 2021; Hessel et al., 2021), Aesthetic Score (Schuhmann et al., 2022), CycleReward (Bahng et al., 2025), HPSv2 (Wu et al., 2023), ImageReward (Xu et al., 2023), Grounded SAM (Ren et al., 2024), and PickScore (Kirstain et al., 2023), along with an ensemble aggregating all eight. We include an oracle setting using the GenEval ground-truth metric as verifier.

As shown in Fig. 11, search with all verifiers consistently improves over the AR baseline across metrics, indicating that *search robustly enhances generation quality*. When ranking verifiers column-wise, each performs best on its own objective. Notably, the *ensemble* typically ranks second on individual metrics but achieves the *best overall average ranking*, demonstrating robust and balanced performance. Among individual verifiers, *ImageReward* and *HPSv2* achieve the strongest average rankings, suggesting that human-preference models serve as effective general verifiers. Full GenEval results and visualizations for different verifiers are provided in App. Table 10 and Figure 21–25 .

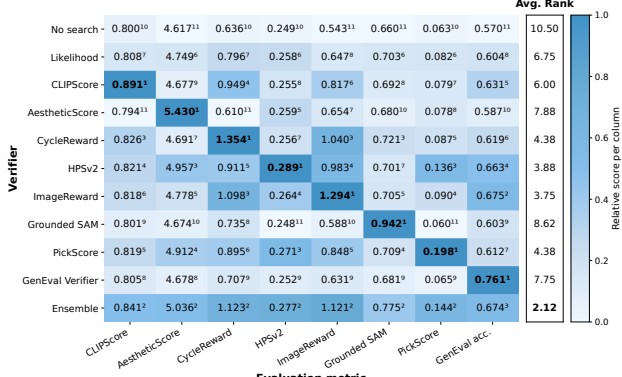

*Figure 11.* **Comparison of different verifiers.** Each row reports search using one verifier. All methods use the same beam search algorithm on FlexTok. The best score in each column is highlighted in bold. The superscript in each cell represents the rank within that column's metric, and the last column reports the average of these column-wise ranks, providing an overall rank for each verifier.

## 6. Related Work

**Image tokenization** Image tokenization creates compressed latent representations of images and enables efficient generative modeling over the latent space, including diffusion (Rombach et al., 2022), masked (Chang et al., 2022), and autoregressive (Ramesh et al., 2021) approaches. The standard approach maps images into a fixed 2D grid of quantized tokens (van den Oord et al., 2017; Esser et al.,

2020), which are typically predicted in raster-scan order by autoregressive generation models (Ramesh et al., 2021; Yu et al., 2022; Sun et al., 2024; Chen et al., 2025b). However, 2D grid tokenization implicitly assumes information is distributed uniformly across space. To overcome this rigidity, 1D tokenizers such as TiTok (Yu et al., 2024) compress images into short 1D sequences (e.g., 32 discrete tokens), greatly reducing the number of tokens required to reconstruct an image. A growing line of work further introduces 1D tokenizers that support variable-length encoding and coarse-to-fine ordering (Yan et al., 2024; Duggal et al., 2025; Miwa et al., 2025; Bachmann et al., 2025; Wen et al., 2025a; Pan et al., 2025; Wang et al., 2025; Liu et al., 2025b), where early tokens capture global structure and later tokens add detail. We refer to this family as *1D ordered tokens*. Among them, FlexTok (Bachmann et al., 2025) achieves stable tokenization down to a single token, produces semantically coherent reconstructions at any prefix length, and the autoregressive models trained on its tokens have shown strong text-to-image generation, making it a natural testbed for our work. Complementary to the advances in tokenization, we study how token structure affects test-time scaling.

**Test-time scaling in image generation** Test-time scaling (a.k.a. inference-time scaling), where additional inference-time compute improves output quality, has proven effective in large language models (Wei et al., 2022; Snell et al., 2024; Huang et al., 2024) and has recently been explored for image generation. For diffusion models, Ma et al. (2025a) established a verifier-plus-search framework, showing that searching over noise trajectories improves generation quality; subsequent works further explore this direction (Singhal et al., 2025; Zhang et al., 2025; He et al., 2025; Sabour et al., 2025). For AR models, TTS methods have been developed for specific token structures: TTS-VAR (Chen et al., 2025c) targets next-scale VAR (Tian et al., 2024) models, while ScalingAR (Chen et al., 2025a) and GridAR (Park et al., 2025) design search strategies for 2D grid-based AR models. Text-based chain-of-thought approaches (Jiang et al., 2025; Guo et al., 2025) offer another direction but require post-training and models with language-generation ability.

In this work, we study how token structure affects test-time scaling, and show that 1D ordered tokens are inherently more amenable to search than 2D grid tokens, even enabling training-free image generation via direct search without an AR model. Several concurrent works provide complementary evidence: Riise et al. (2025) show that next-scale AR models scale better at test time than diffusion models; Self-Tok (Wang et al., 2025) finds that RL-based post-training on ordered tokens yields larger gains than on 2D grid tokens; and Beyer et al. (2025) demonstrate training-free generation from highly compressed 1D tokens. Different from these works, we systematically study how token structure inter-

acts with search algorithms, verifiers, and AR priors under controlled settings to provide a holistic view.

Please refer to the Appendix A for additional related work.

# 7. Conclusion and Limitations

In this work, we investigated how token structure influences the effectiveness of test-time search in autoregressive image generation. We showed that 1D ordered tokenizers with a coarse-to-fine structure are inherently more amenable to search, and empirically demonstrated their advantages in test-time scaling. Beyond improved scaling behavior, we found that this structure enables effective zero-shot control, and, in the extreme, allows training-free image generation via direct search over the ordered token space. We further provided a systematic analysis of how search strategies, verifiers, and autoregressive priors interact with token structure. We discuss limitations of our study and future work below, with additional failure case analysis in App. G.

**Search algorithms.** In this work, search is primarily used as a diagnostic tool to study token structure, rather than as a fully optimized component. The search strategies evaluated are largely generic and not tailored to the structure of ordered tokens. Designing search algorithms that explicitly exploit the ordered coarse-to-fine hierarchy, such as deciding which token positions to search, and performing more effective branching (or early stopping), could further improve both efficiency and generation quality.

**Verifiers.** While we evaluate a diverse set of verifiers and show consistent improvements from search, generation quality is fundamentally bounded by verifier reliability. With sufficient compute, search can exploit verifier weaknesses (i.e., verifier hacking). Besides, most verifiers provide only global scalar feedback, limiting fine-grained corrections. Developing verifiers robust to such exploitation and offering dense, spatially-localized feedback remains a key challenge.

**Detokenization overhead.** Our experiments focus on a 1D ordered tokenizer with a flow-based detokenizer that requires multiple denoising steps. This introduces a computational bottleneck during search, as intermediate decoding is repeatedly required for verification (cf. Fig. 14). Future work could explore more efficient decoding mechanisms (e.g., one-step detokenization), adaptive decoding schedules, or latent-space verifiers that avoid detokenization entirely.

**Generality across generation paradigms.** Our results suggest that image tokenization is not solely a representational choice but also a mechanism for effective test-time search. Future research can design tokenization, or more generally generation paradigms, good for test-time scaling. Beyond image generation, we expect to extend the findings to other domains, such as text, video, or multimodal generation.

## Acknowledgment

We thank Ali Garjani and Jiachen Lu for constructive discussions and assistance in preparing the manuscript. We are also grateful to Muhammad Uzair Khattak, Mingfei Gao, and Anders Boesen Lindbo Larsen for their valuable feedback on earlier versions of the manuscript. We further thank Yizhou Xu and Zhekai Jiang for helpful discussions on the theoretical aspects of this work, as well as Vikhyat Agrawal and Yuanzhong Chen for feedback on the codebase and its extensions. We acknowledge Lambda for supporting this paper through their academic compute grant program, and Apple for a gift. This work was supported under project ID 43 as part of the Swiss AI Initiative, through a grant from the ETH Domain, with computational resources provided by the Swiss National Supercomputing Centre (CSCS) on the Alps infrastructure. This work has also received funding from the Swiss State Secretariat for Education, Research and Innovation (SERI).

## Impact Statement

This work studies how token structure affects test-time search in autoregressive image generation, with the goal of improving the controllability and quality of generated images. As with image-generation methods more broadly, these capabilities may also be misused to produce misleading or deceptive imagery, and generated outputs may reflect biases present in the underlying models' training data. Our work does not introduce a new generative model or training dataset, and the risks we discuss are primarily associated with the pretrained generative models used in our experiments. Nevertheless, improved understanding of test-time search may affect how such systems are controlled and deployed, so these risks should be considered in practice.

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

# Table of Contents

# A. Additional Related Work

In this section, we summarize additional related work relevant to our study, providing a broader context for our paper.

**Search in AI and LLMs**   Search has long been a cornerstone of classical AI, with algorithms such as minimax, alpha–beta pruning, and Monte Carlo Tree Search (MCTS) (Coulom, 2006; Browne et al., 2012) achieving strong performance in domains like chess, Go, and planning (Russell & Norvig, 2010; Campbell et al., 2002). Modern models such as AlphaGo (Silver et al., 2016; 2017) and AlphaZero (Silver et al., 2018) show that combining learned priors with search yields superhuman performance and benefits from more test-time compute, a pattern also seen in poker (Brown & Sandholm, 2018). Search has become increasingly important in large language models (LLMs), particularly for complex reasoning tasks (Wang et al., 2022; Yao et al., 2023; Lightman et al., 2023), where search is performed over intermediate thinking steps. Recent studies have primarily focused on developing better search algorithms (Yao et al., 2023; Snell et al., 2024; Besta et al., 2024), designing stronger verifiers for evaluating intermediate or final reasoning steps (Lightman et al., 2023; Zhu et al., 2023; Wang et al., 2024), and understanding the scaling laws of test-time compute (Snell et al., 2024; Brown et al., 2024; Wu et al., 2024b).

Compared to these domains, intermediate token sequences in image generation typically lack clear semantic meaning, and the flat token space does not naturally lend itself to efficient search strategies. Our work shows that 1D ordered tokenizers can help address these challenges by producing semantically interpretable intermediate states with a coarse-to-fine structure, bringing image generation closer to domains where search has proven effective.

**RL-based training for image generation**   A growing body of work applies reinforcement learning (RL) to improve image generation using similar reward signals as those that serve as verifiers in test-time search. DDPO (Black et al., 2023) formulates diffusion denoising as a multi-step Markov Decision Process and applies policy gradients to fine-tune on downstream objectives. More recently, GRPO-based methods (Shao et al., 2024) have been adapted from LLMs to visual generation: DanceGRPO (Xue et al., 2025) and Flow-GRPO (Liu et al., 2025a) adapt GRPO to diffusion and flow-based models, AR-GRPO (Yuan et al., 2025) applies GRPO to next-token-prediction AR image generators, and Gallici & Borde (2025) integrate GRPO with next-scale VAR models. A separate line of work incorporates textual chain-of-thought reasoning into image generation via RL (Guo et al., 2025; Jiang et al., 2025), enabling models to plan in language before producing visual tokens. Notably, most RL-finetuned image generation models still operate with fixed inference-time compute in the visual generation process itself. Where additional compute is introduced, it is typically through language-based reasoning rather than in the image token space. Our work studies test-time scaling by search directly over the image token space.

*Connection to search.* RL and search both steer generation toward higher-quality outputs guided by a reward/verifier, but RL reshapes the model's distribution through training with a fixed objective, while search explores per instance at inference time, offering flexibility to target different objectives by swapping verifiers without retraining. Empirically, these two approaches have been found to be complementary in both LLMs (Snell et al., 2024; Huang et al., 2024; Yue et al., 2025) and diffusion models (Ma et al., 2025a), where search can further improve already reward-finetuned models. Our work focuses on the search axis, and can be applied on top of RL-finetuned models as well.

**Controllability and verifiers in image generation**   Controllability is a long-standing goal in generative modeling, pursued through diverse approaches. Classifier guidance (Dhariwal & Nichol, 2021) and classifier-free guidance (CFG) (Ho & Salimans, 2022) strengthen conditional generation by amplifying conditioning signals during inference. Spatial conditioning methods such as ControlNet (Zhang et al., 2023) inject structural signals (e.g., edges, depth, pose) into pretrained models, enabling fine-grained layout control. Preference-based learning with reward models such as ImageReward (Xu et al., 2023), HPSv2 (Wu et al., 2023), PickScore (Kirstain et al., 2023), and CycleReward (Bahng et al., 2025) improves text-image alignment and aesthetic quality through RL-based finetuning.

Our work offers an orthogonal angle, improving controllability through verifier-guided search at inference time. By swapping verifiers, one can steer generation toward different objectives on the fly without retraining. Our framework naturally benefits from advances in other areas, such as stronger AR models that narrow the search space and better reward models that provide more reliable verification signals.

**Training-time and test-time compute**   The performance of generative models can be improved by scaling compute along two axes: training-time compute, by increasing model size, data, and training duration (Kaplan et al., 2020; Hoffmann et al., 2022; Zhai et al., 2022; Peebles & Xie, 2023), and test-time compute, by allocating additional computation during

inference (Snell et al., 2024; Muennighoff et al., 2025; Ma et al., 2025a). The tradeoff between these two axes has been studied across domains. In board games, combining learned priors with search yields superhuman performance that improves with more test-time compute (Silver et al., 2016; Brown & Sandholm, 2018), and Jones (2021) explicitly characterize how training-time and test-time compute can be exchanged using MCTS on the game of Hex. For LLMs, Snell et al. (2024) show that with compute-optimal allocation, a smaller model can outperform one $14\times$ larger on math reasoning. For diffusion-based image generation, Ma et al. (2025a) show that a smaller diffusion model with more test-time compute can achieve similar performance to a larger model.

Our work takes a step in this direction by studying how model size and prior strength interact with test-time compute in autoregressive image generation. We find that a smaller AR model with sufficient test-time search can outperform a larger one without search, and that even with minimal training-time compute (e.g., no trained AR model), sufficient test-time search over ordered tokens can produce reasonable images.

**Ordered tokenization beyond images** The coarse-to-fine ordered tokenization we study has recently been extended to several other modalities. VideoFlexTok (Atanov et al., 2026) brings flexible-length, coarse-to-fine tokenization to video, where early tokens capture semantics and motion and later tokens add detail. In 3D, LoST (Dutt et al., 2026) orders tokens by semantic salience so that short prefixes already decode into complete, plausible shapes. In robotics, OAT (Liu et al., 2026) learns an ordered, prefix-decodable action representation that trades inference cost for action fidelity. In biology, Dilip et al. (2026) tokenize protein structures such that successive tokens add increasing detail to a global representation. These works indicate that coarse-to-fine ordered tokenization generalizes well beyond images, making these modalities natural settings to explore test-time search in future work.

## B. Theoretical Analysis

**Intuition.** Searching over image tokens to satisfy a target property (e.g., image–text alignment) can be viewed as a structured retrieval problem: the goal is to identify an image configuration that maximizes a verifier score within a large combinatorial space. The efficiency of this search depends critically on how the token space is organized. This is analogous to classical search data structures, where organizing data along informative directions (e.g., kd-trees (Bentley, 1975) or PCA-based partitioning (McNames, 2002)) can significantly accelerate retrieval. Both 1D ordered tokenizers and 2D grid tokenizers impose structure over image representations, but in fundamentally different ways. A 1D ordered tokenizer arranges tokens according to their information contribution, so that early tokens capture globally salient content that is most relevant to a semantic verifier. In contrast, a 2D grid tokenizer organizes tokens according to spatial locality, without explicitly aligning token order with semantic importance. Intuitively, a representation that prioritizes information relevant to the objective should enable more effective and reliable search.

We formalize this intuition by analyzing the relationship between the search gap (i.e., the difference between the optimal verifier score and the score achieved by search) and the structure of the token space. We first show that the search gap is bounded by the heuristic error (Pearl, 1983), which measures how accurately intermediate partial generations predict the final verifier score (Proposition B.1). We then connect this heuristic error to the reconstruction error between intermediate decoded images and their possible full completions under a Lipschitz assumption on the verifier (Proposition B.2). This establishes that search performance depends on how well intermediate states approximate the final image. Since 1D ordered tokenizers are trained with nested dropout to explicitly minimize this intermediate reconstruction error, they induce tighter bounds on the search gap compared to 2D grid tokenizers.

**Setup.** Let $x^* = \arg\max_x g(x, c)$ be the optimal image for a given verifier $g$ and context $c$, and let $\hat{x}$ denote the image found by a heuristic search algorithm. Define the search gap $\Delta = g(x^*, c) - g(\hat{x}, c) \geq 0$. Let $x_{1:t}$ denote the decoded intermediate image obtained from a token prefix via a partial decoder, and define the optimal continuation value $F_t(x_{1:t}) := \max_{x_{t+1:T}} g(x_{1:T}, c)$. The heuristic error is $B_t := \sup_{x_{1:t}} |F_t(x_{1:t}) - g(x_{1:t}, c)|$. Let $t_0$ be the critical step at which the search algorithm permanently prunes the optimal prefix $x^*_{1:t_0}$, and define the continuation suboptimality $\eta_{t_0} := F_{t_0}(\hat{x}_{1:t_0}) - g(\hat{x}, c) \geq 0$.

**Proposition B.1.** *The search gap satisfies*
$$\Delta \leq 2B_{t_0} + \eta_{t_0}.$$

*Proof sketch.* Assume the algorithm diverges at step $t_0$, preferring $\hat{x}_{1:t_0}$ over $x^*_{1:t_0}$ (i.e., $g(\hat{x}_{1:t_0}, c) \geq g(x^*_{1:t_0}, c)$). By the

definition of $B_t$, we have

$$F_{t_0}(x^*_{1:t_0}) \le g(x^*_{1:t_0}, c) + B_{t_0} \le g(\hat{x}_{1:t_0}, c) + B_{t_0} \le F_{t_0}(\hat{x}_{1:t_0}) + 2B_{t_0}.$$

Since $F_{t_0}(x^*_{1:t_0}) = g(x^*, c)$ and $g(\hat{x}, c) = F_{t_0}(\hat{x}_{1:t_0}) - \eta_{t_0}$, we obtain $\Delta \le 2B_{t_0} + \eta_{t_0}$. (Note: The AR prior can be incorporated by replacing $g(x, c)$ with $g(x, c) + \lambda \log p_\omega(x)$.) □

**Proposition B.2.** *Assume the verifier $g$ is $L$-Lipschitz with respect to the decoded image space, and the reconstruction error between intermediate tokens and all consistent completions is bounded by $\epsilon_t$, i.e.,*

$$\sup_{x \in \mathcal{C}(x_{1:t})} \|x_{1:t} - x\|_2 \le \epsilon_t.$$

*Then the heuristic error satisfies $B_t \le L\epsilon_t$.*

*Proof sketch.* For any $L$-Lipschitz verifier $g$, and any completion $x \in \mathcal{C}(x_{1:t})$, we have $|g(x_{1:t}, c) - g(x, c)| \le L\|x_{1:t} - x\|_2$. Taking supremum over completions and prefixes gives $B_t \le L\epsilon_t$. □

**Combined bound.** Substituting Proposition B.2 into Proposition B.1, we obtain

$$\Delta \le 2L\epsilon_{t_0} + \eta_{t_0},$$

which directly links the search gap to the reconstruction discrepancy $\epsilon_{t_0}$ of the tokenizer at the critical pruning step. Here, $\eta_{t_0}$ captures the suboptimality of the search after the divergence point $t_0$. Under a sufficiently strong search procedure (e.g., large-beam search), $\eta_{t_0}$ is typically small, as the continuation from $\hat{x}_{1:t_0}$ approximately maximizes $g$. Moreover, it vanishes under early stopping, where the intermediate decode $x_{1:t_0}$ is returned directly, yielding $\Delta \le 2L\epsilon_{t_0}$. This is particularly natural for 1D ordered tokenizers, whose intermediate decodes are semantically meaningful due to nested dropout.

- **1D Ordered Tokens:** 1D Ordered Tokens explicitly minimize intermediate reconstruction error via nested dropout ($\mathcal{L}_{\text{nested}} = \mathbb{E}_t[\|x_{1:t} - x\|_2^2]$), encouraging $\epsilon_t$ to remain small throughout generation. Under a linear reconstruction assumption, this relates to a PCA-like decomposition where $\epsilon_t = \left(\sum_{s=t+1}^T \lambda_s^2\right)^{1/2}$ (Rippel et al., 2014). Because leading components capture dominant global variance early, $\epsilon_t$ decreases rapidly with $t$.

- **2D Grid Tokens:** For spatial grid tokenizations, the reconstruction objective is only enforced at $t = T$. At intermediate steps ($t \ll T$), large portions of the image remain unconstrained, leading to potentially large $\epsilon_t$ and thus loose heuristic bounds.

**Discussion.** Our analysis demonstrates that the search gap $\Delta$ is fundamentally governed by the reconstruction error of intermediate images. 1D ordered tokens explicitly minimize this error via nested dropout, maintaining a progressively tightening bound that provides reliable guidance from the earliest tokens. In contrast, 2D grid tokens offer no such structural guarantee, leading to weaker heuristic guidance at early stages. While lookahead rollouts can partly mitigate the heuristic error for 2D grid tokens, they come at substantially higher computational cost and are less effective in settings with weak priors, such as uniform priors or zero-shot multimodal control. These theoretical findings are consistent with our empirical results in Section 5.

# C. Method and Implementation Details

This section provides additional methods and implementation details for the search algorithms and verifiers explored.

## C.1. Search Algorithms

Below, we provide detailed formulations and pseudocode for the three search algorithms used in this paper.

As defined in Sec. 2 of the main paper, an image is represented as a sequence of $T$ discrete tokens

$$\mathbf{x} = (x_1, x_2, \ldots, x_T), \qquad x_t \in \mathcal{V},$$

where $\mathcal{V}$ is the token vocabulary. We assume a verifier function $g : \mathcal{V}^T \to \mathbb{R}$, which assigns a scalar score to a (possibly decoded) image. For clarity, we here also define a *next-token prior* model $p(x_t \mid x_{<t})$, typically an autoregressive (AR) image model conditioned on a text prompt (omitted in notation for simplicity). When such a prior model is unavailable, we also allow a uniform next-token prior (as in Sec. 3.2). Since most verifiers operate on images rather than raw tokens, we denote by $\mathrm{Dec}(\cdot)$ the detokenizer that converts a token sequence into an image. For *image-based verifiers*, we therefore write

$$g(\mathbf{x}) = g_{\mathrm{img}}(\mathrm{Dec}(\mathbf{x})).$$

throughout this section. We note, however, that some verifiers operate directly on token sequences (e.g., likelihood-based verifiers); these can be written as $g_{\mathrm{tok}}(\mathbf{x})$ and do not require detokenization. For conciseness, the descriptions below focus on the image-based case, but all algorithms apply to token-based verifiers as well.

**Best-of-$N$ sampling.** Best-of-$N$ sampling draws $N$ independent sequences from the next-token prior model and selects the one with the highest verifier score. Formally,

$$\mathbf{x}^{\star} = \arg \max_{i \in \{1,\ldots,N\}} g_{\mathrm{img}}\Big(\mathrm{Dec}(\mathbf{x}^{(i)})\Big),$$

where each sequence $\mathbf{x}^{(i)} = (x_1^{(i)}, \ldots, x_T^{(i)})$ is generated autoregressively via $x_t^{(i)} \sim p(x_t \mid x_{<t})$.

We note that Best-of-$N$ relies crucially on an informative next-token prior; with a uniform prior, all $N$ samples are effectively random trajectories in the full token space and thus cannot produce meaningful images. We provide pseudocode in Algorithm 1.

---

**Algorithm 1** Best-of-$N$ Sampling

---

**Require:** Next-token prior $p(\cdot \mid \cdot)$, verifier $g_{\mathrm{img}}$, detokenizer $\mathrm{Dec}(\cdot)$, number of samples $N$
**Ensure:** Generated image
1: **for** $i = 1$ to $N$ **do**
2:      $\mathbf{x} \leftarrow []$
3:      **for** $t = 1$ to $T$ **do**
4:          Sample $x_t \sim p(x_t \mid \mathbf{x})$
5:          Append $x_t$ to $\mathbf{x}$
6:      **end for**
7:      $\mathrm{img}[i] \leftarrow \mathrm{Dec}(\mathbf{x})$
8:      $\mathrm{score}[i] \leftarrow g_{\mathrm{img}}(\mathrm{img}[i])$
9: **end for**
10: $i^{\star} \leftarrow \arg \max_i \mathrm{score}[i]$
11: **return** $\mathrm{img}[i^{\star}]$

---

**Beam search.** Beam search is a guided tree search where each node (partial sequence) expands to a few likely next tokens, and only the most promising branches are kept at every step. This allows the verifier to guide generation throughout the process, rather than only evaluating complete images.

Specifically, beam search alternates between (1) expanding each prefix using the next-token prior and (2) selecting the top-$k$ prefixes according to the verifier. Given a beam $B_{t-1}$ at step $t-1$, we first obtain, for each prefix $\mathbf{x}_{1:t-1} \in B_{t-1}$, a set of $M$ candidate next tokens sampled from the conditional prior $p(x_t \mid \mathbf{x})$. We then construct the expanded candidate set

$$\mathrm{Cand}_t = \big\{\mathbf{x}_{1:t-1} \circ x_t \mid \mathbf{x}_{1:t-1} \in B_{t-1},\ x_t \in \mathcal{N}_t(\mathbf{x}_{1:t-1})\big\},$$

where $\circ$ denotes token concatenation. Among these $kM$ expanded prefixes, we retain the $k$ highest-scoring ones based on partial-image verifier scores:

$$B_t = \mathrm{Top}_k(g_{\mathrm{img}}(\mathrm{Dec}_{\mathrm{partial}}(\mathbf{x}_{1:t})) :\ \mathbf{x}_{1:t} \in \mathrm{Cand}_t).$$

After $T$ steps, beam search selects the completed sequence with the highest full-image score:

$$\mathbf{x}^{\star} = \arg \max_{\mathbf{x} \in B_T} g_{\mathrm{img}}(\mathrm{Dec}(\mathbf{x})).$$

**Importantly**, the partial detokenizer $\text{Dec}_{\text{partial}}$ depends on *the underlying token structure.* For 1D ordered token sequences such as FlexTok (Bachmann et al., 2025), we can directly detokenize the current prefix into an image for verification. In contrast, for 2D grid tokenizations used in Janus (Wu et al., 2024a; Chen et al., 2025b), we obtain a partial image by padding the ungenerated grid locations with zeros.

More generally, *beam search does not need to be applied at every token step.* Instead, we can evaluate the verifier only at sparse token positions. In this variant, the autoregressive model is rolled forward for $k$ steps before a verification and selection step is performed (e.g., applying search only at token indices $64, 128, 256, \ldots$). Formally, for a skip length $s$, the candidate set becomes

$$\mathcal{N}_{t+s}(\mathbf{x}_{1:t}) = \{x_{t+s}^{(1)}, \ldots, x_{t+s}^{(M)} \mid x_{t+s}^{(i)} \sim p(x_{t+s} \mid \mathbf{x}_{1:t})\}.$$

Each candidate $x_{t+s}^{(i)}$ corresponds to rolling the AR model forward $s$ steps without verifier guidance and applying selection only at the $(t+s)$-th token.

Because beam search applies verification and guidance during the generation path, it can still produce meaningful images even when no informative prior model is available (as shown in Sec. 3.2). Pseudocode for beam search is provided in Algorithm 2.

---

**Algorithm 2** Beam Search (with Sparse Verification)

---

**Require:** Next-token prior $p(\cdot \mid \cdot)$, verifier $g_{\text{img}}$, detokenizer $\text{Dec}_{\text{partial}}$ (or Dec), beam size $k$, width $M$, skip length $s$
**Ensure:** Best generated image
 1: $B \leftarrow \{\text{empty sequence}\}$
 2: **for** $t = 1$ to $T$ **do**
 3:     $\text{Cand} \leftarrow []$
 4:     **if** $t \bmod s = 0$ **then**                              ▷ search step
 5:         **for** each prefix $\mathbf{x}$ in $B$ **do**
 6:             Sample $\{x_t^{(1)}, \ldots, x_t^{(M)}\} \sim p(x_t \mid \mathbf{x})$
 7:             **for** each $x_t^{(i)}$ **do**
 8:                 $\mathbf{x}' \leftarrow \mathbf{x} \circ x_t^{(i)}$
 9:                 $\text{img} \leftarrow \text{Dec}_{\text{partial}}(\mathbf{x}')$
10:                 $v \leftarrow g_{\text{img}}(\text{img})$
11:                 Append $(\mathbf{x}', v)$ to Cand
12:             **end for**
13:         **end for**
14:         $B \leftarrow \{\mathbf{x}' \mid (\mathbf{x}', v) \in \text{Top}_k(\text{Cand})\}$
15:     **else**                                          ▷ roll forward without branching
16:         **for** each prefix $\mathbf{x}$ in $B$ **do**
17:             Sample a single token $x_t^r \sim p(x_t \mid \mathbf{x})$
18:             $\mathbf{x}' \leftarrow \mathbf{x} \circ x_t^r$
19:             Append $\mathbf{x}'$ to Cand
20:         **end for**
21:         $B \leftarrow \text{Cand}$
22:     **end if**
23: **end for**
24: $\mathbf{x}^\star \leftarrow \arg\max_{\mathbf{x} \in B} g_{\text{img}}(\text{Dec}(\mathbf{x}))$
25: **return** $\text{Dec}(\mathbf{x}^\star)$

---

**Lookahead search.** Lookahead search follows the same procedure as beam search but replaces partial-image verification with a rollout-based evaluation. Instead of detokenizing the current prefix $\mathbf{x}_{1:t}$ directly, lookahead rolls out $L$ additional tokens using the next-token prior before calling the verifier. This provides the verifier with more complete visual context, especially when early tokens contain little semantic information (e.g., in 2D grid tokenizations).

Formally, lookahead follows the same procedure as beam search except in the scoring stage. Instead of directly detokenizing the partial prefix with $\text{Dec}_{\text{partial}}$, lookahead replaces this with a rollout-based detokenization: each prefix is evaluated only

after rolling it out for $L$ additional autoregressive steps. The selection step therefore becomes

$$B_t = \mathrm{Top}_k(g_{\mathrm{img}}(\mathrm{Dec}(\mathbf{x}_{1:t} \circ \mathbf{x}_{\mathrm{la}})) : \mathbf{x}_{1:t} \in \mathrm{Cand}_t),$$

where

$$\mathbf{x}_{\mathrm{la}} = (x_{t+1}, \dots, x_{t+L}),$$

and the rollout tokens are sampled from the next-token prior, $x_{t+\ell} \sim p(x_{t+\ell} \mid \mathbf{x}_{1:t+\ell-1})$ for $\ell \in [1, L]$.

Note that only the prefix $\mathbf{x}_{1:t}$ is retained in the candidate set and beam; the rollout tokens are used solely for scoring and discarded afterward. We provide pseudocode in Algorithm 3, with the differences from beam search highlighted in blue for clarity.

---

**Algorithm 3** Lookahead Search (differences from Beam Search in blue)

---

**Require:** Next-token prior $p(\cdot \mid \cdot)$, verifier $g_{\mathrm{img}}$, detokenizer Dec, beam size $k$, width $M$, skip length $s$, rollout length $L$
**Ensure:** Best generated image
1: $B \leftarrow \{\text{empty sequence}\}$
2: **for** $t = 1$ to $T$ **do**
3:     Cand $\leftarrow []$
4:     **if** $t \bmod s = 0$ **then**                          ▷ search step
5:         **for** each prefix $\mathbf{x}$ in $B$ **do**
6:             Sample $\{x_t^{(1)}, \dots, x_t^{(M)}\} \sim p(x_t \mid \mathbf{x})$
7:             **for** each $x_t^{(i)}$ **do**
8:                 $\mathbf{x}' \leftarrow \mathbf{x} \circ x_t^{(i)}$
9:                 Initialize rollout token list: $\mathbf{x}_{\mathrm{la}} \leftarrow []$
10:                **for** $\ell = 1$ to $L$ **do**
11:                    Sample $x_{t+\ell} \sim p(x_{t+\ell} \mid \mathbf{x}' \circ \mathbf{x}_{\mathrm{la}})$
12:                    Append $x_{t+\ell}$ to $\mathbf{x}_{\mathrm{la}}$
13:                **end for**
14:                 img $\leftarrow \mathrm{Dec}(\mathbf{x}' \circ \mathbf{x}_{\mathrm{la}})$
15:                 $v \leftarrow g_{\mathrm{img}}(\text{img})$
16:                 Append $(\mathbf{x}', v)$ to Cand
17:             **end for**
18:         **end for**
19:         $B \leftarrow \{\mathbf{x}' \mid (\mathbf{x}', v) \in \mathrm{Top}_k(\mathrm{Cand})\}$
20:     **else**                            ▷ roll forward without branching
21:         **for** each prefix $\mathbf{x}$ in $B$ **do**
22:             Sample a single token $x_t^r \sim p(x_t \mid \mathbf{x})$
23:             $\mathbf{x}' \leftarrow \mathbf{x} \circ x_t^r$
24:             Append $\mathbf{x}'$ to Cand
25:         **end for**
26:         $B \leftarrow$ Cand
27:     **end if**
28: **end for**
29: $\mathbf{x}^\star \leftarrow \arg\max_{\mathbf{x} \in B} g_{\mathrm{img}}(\mathrm{Dec}(\mathbf{x}))$
30: **return** $\mathrm{Dec}(\mathbf{x}^\star)$

---

## C.2. Verifiers

Verifiers serve as the search objective, providing guidance for both partial and complete token sequences. As in the main paper, we group the verifiers into three major categories: (1) *image–text alignment* verifiers, which include CLIPScore, ImageReward, HPSv2, PickScore, CycleReward, likelihood-based scoring, and our rule-based verifier; (2) *image–image alignment* verifiers, represented by DreamSim; (3) *image-quality* verifiers, such as the LAION Aesthetic Score; and (4) an *ensemble* of them. We summarize the verifiers in Table 3 and describe implementation details for each verifier below.

| Category | Verifier | Description / Purpose |
|---|---|---|
| **Image–text alignment** | CLIPScore (Radford et al., 2021; Hessel et al., 2021) | Contrastive image–text similarity using CLIP ViT-B/32; fast global semantic alignment. |
| | ImageReward (Xu et al., 2023) | Reward model trained on human comparisons; strong on semantic correctness and perceptual realism. |
| | PickScore (Kirstain et al., 2023) | Trained on user comparisons; lightweight and effective for global semantics and composition. |
| | HPSv2 (Wu et al., 2023) | Large unified human-preference model; strong on fine-grained quality, realism, and adherence. |
| | CycleReward (Combo) (Bahng et al., 2025) | Combines cycle-consistency and preference-based objectives; stable semantic alignment. |
| | Likelihood | Token-level self-likelihood $\sum \log p(x_i \mid x_{<i})$; reflects model fluency without detokenization. |
| | Rule-based | Object grounding + segmentation (via Grounded SAM (Ren et al., 2024)) for checking object existence, count, color, and spatial relations. |
| **Image–image alignment** | DreamSim (Fu et al., 2023) | Perceptual similarity model trained on human triplets; strong for reference-image guidance. |
| **Image quality** | LAION Aesthetic Score (Schuhmann et al., 2022) | Predicts aesthetic appeal, composition, clarity; complements semantic verifiers. |
| **Ensemble** | Rank-based fusion (Ma et al., 2025a) | Aggregates ranks across verifiers; robust to scale differences and leverages complementary strengths. |

*Table 3.* Summary of all verifiers used in our work.

**CLIPScore.** We use OpenAI CLIP ViT-B/32 (Radford et al., 2021) and compute the CLIPScore as defined in Hessel et al. (2021):

$$\text{CLIPScore} = 2.5 \times \max(\text{cos\_sim}, 0),$$

where cos_sim is the cosine similarity between normalized CLIP embeddings of the text prompt and generated image. CLIPScore provides a fast semantic alignment signal.

**ImageReward.** ImageReward (Xu et al., 2023) is trained on 137K human image–prompt preference pairs. It predicts which image better matches human intent, capturing semantic correctness, object quality, and perceptual realism. We use the official ImageReward-v1 checkpoint.

**PickScore.** PickScore (Kirstain et al., 2023) is trained on the Pick-a-Pic dataset, which contains real user preference comparisons gathered from an online image-generation interface. It is lightweight and performs well for global semantic alignment and composition quality.

**HPSv2.** HPSv2 (Wu et al., 2023) is a large-scale reward model trained on a unified mixture of human preference datasets. Compared to ImageReward and PickScore, HPSv2 more reliably captures fine-grained prompt adherence and style consistency.

**CycleReward (Combo).** CycleReward (Bahng et al., 2025) introduces a self-supervised cycle-consistency objective between text and image embeddings without relying directly on human annotations. The *Combo* variant aggregates multiple reward heads (e.g., cycle-based and preference-based), producing a stable alignment score and improving robustness across prompt types.

**Likelihood-based verifier.** For autoregressive models with accessible token probabilities, we compute the token-level self-likelihood

$$g_{\text{tok}}(x_{1:t}) = \sum_{i=1}^{t} \log p(x_i \mid x_{<i}),$$

which reflects the model's internal image-text consistency. This score requires no detokenization and is therefore efficient to evaluate. However, it is inherently limited by the predictive capability and biases of the AR model itself, and in practice tends to yield only limited improvements in image quality or alignment.

**Rule-based verifier (Grounded SAM).**   To evaluate fine-grained, structured constraints such as object presence, count, color, and spatial relations, we implement a rule-based verifier built on Grounded SAM (Ren et al., 2024). Our design follows the GenEval (Ghosh et al., 2023) evaluation pipeline, but replaces the Mask2Former detector with an open-vocabulary segmentation pipeline (GroundingDINO + SAM) to support more general prompts, and replaces binary spatial checks with a continuous scoring scheme for improved robustness.

Given an object phrase from the prompt (e.g., "a red apple", "a cat", "a dog to the left of a chair"), GroundingDINO predicts text-conditioned bounding boxes and SAM produces corresponding segmentation masks. From these masks, the verifier computes: (i) *object existence and count* by matching detected masks to the specified object; (ii) *color consistency* using CLIP-based classification on cropped object regions, following Ghosh et al. (2023); and (iii) *spatial relation accuracy*, obtained by comparing object masks along the axis implied by the relation (e.g., "left of", "in front of"). This is computed by projecting mask centroids or support regions onto the relevant axis and converting the relative ordering into a continuous score in $[0, 1]$ (Rezaei et al., 2025), yielding a smooth and stable spatial signal instead of a brittle pass/fail check. All criteria are aggregated into a single continuous score in $[0, 1]$, allowing the rule-based verifier to provide interpretable and localized guidance to the search algorithm.

Note that this verifier requires parsing the prompt into structured attributes (objects, colors, relations). In our experiments on GenEval, we use the provided metadata directly; for general usage, this parsing would typically require an LLM or VLM to extract the necessary attributes from free-form text.

**DreamSim.**   DreamSim (Fu et al., 2023) is a perceptual similarity model trained on human-labeled triplets. It provides a strong reference-image alignment signal, capturing fine-grained textures and semantics similarity accurately.

**Aesthetic Score.**   We use the LAION aesthetic predictor (Schuhmann et al., 2022), trained on human-rated aesthetic labels. It produces a continuous score reflecting visual appeal, clarity, composition, and style. This verifier complements semantic scores by penalizing visually low-quality outputs.

**Ensemble verifiers.**   Following Ma et al. (2025a), we combine multiple verifiers using rank-based aggregation. Each candidate is ranked independently by each verifier, and the ranks are summed (or averaged) to produce an aggregated score. This avoids inconsistencies between heterogeneous scoring scales and leverages the complementary strengths of different verifiers, yielding more robust guidance during search.

### C.3. Tokenizer and AR Models

We use pretrained tokenizers and autoregressive image generation models across all experiments, including FlexTok (Bachmann et al., 2025), Janus (Wu et al., 2024a), and a 2D grid tok variant from FlexTok (used as a tokenization ablation; see the FlexTok paper (Bachmann et al., 2025) for details). Each model is paired with its official tokenizer and publicly released AR checkpoint. For FlexTok, we use the largest 3.4B AR model as the default and additionally evaluate other released sizes (212M, 530M, 1.4B, 3.4B) for scaling analysis.

All AR models are trained for text-to-image generation. For unconditional AR experiments, we run FlexTok with an empty prompt (i.e., the CFG token only). Across all models, we keep their official sampling hyperparameters (e.g., temperature, top-$k$, top-$p$, classifier-free guidance) exactly as released and do not retune any sampling settings.

## D. Experiment Settings

This section provides the dataset protocols, evaluation settings, search configurations, and inference-time compute metrics used throughout our experiments.

### D.1. Dataset and Evaluation Settings

We evaluate on three benchmarks covering both text-to-image and reference-guided generation: GenEval, COCO Captions, and DreamBench++. We describe each benchmark and the evaluation protocol below.

**GenEval**   (Ghosh et al., 2023) GenEval measures compositional text-to-image alignment with respect to object presence, object count, color attributes, and spatial relations. It contains 553 prompts, each describing a simple but compositionally challenging scene (e.g., "A blue cup on the left of a pink table"). The evaluation uses a rule-based verification pipeline using multiple models like Mask2Former and CLIP (Radford et al., 2021). We follow the common practice of using the average of 5 examples for evaluation, and run the official evaluation protocol to get the results.

**MS-COCO**   (Lin et al., 2014) MS-COCO evaluates general text-to-image generation quality. We use a subset of 300 captions from the MS-COCO validation captions (Karpathy split). It covers a broad range of everyday scenes and realistic photographic styles. We use the original captions without augmentation and report CLIP-based image–text alignment scores as the primary metric, while also including other verifiers as supplemental evaluations.

**DreamBench++.**   (Peng et al., 2024) DreamBench++ evaluates concept preservation and reference-guided generation. It contains 1,350 instances, each consisting of a text prompt paired with a reference image. Generated images are evaluated using CLIP and DINO similarities with respect to the reference, measuring both image-text semantic consistency and image-image consistency. DreamBench++ serves as a benchmark for scenarios requiring multiple forms of control.

### D.2. Search Configuration

Unless otherwise specified, we use a consistent set of hyperparameters for each search algorithm across all experiments. *For beam search*, we use a beam width of $k = 5$ and a candidate width of $M = 10$ for all AR models. *Lookahead search* uses the same $(k, M)$ configuration, and unless otherwise specified, the rollout continues until the end of the sequence. For *Best-of-N*, we vary $N$ from 1 to 50.

To control the number of search steps in beam search and lookahead search, we adopt different verification schedules depending on the tokenizer. For FlexTok, we typically use exponentially spaced steps, e.g., $t \in \{2^0, 2^1, 2^2, \dots, 2^8\}$ (9 steps), following its exponential training schedule and as used in the verifier analysis experiments. For **2D grid tokens** (including Janus), verification is instead performed at uniformly spaced positions, e.g., $t = 32 \times n$ for $n = 0, \dots, 8$ (9 steps). In our controlled comparisons, we also apply uniform verification to FlexTok for fairness. However, in general, exponential spacing yields better performance. Designing more effective verification schedules remains an open direction for future work.

### D.3. Inference-Time Compute

Following prior work (Ma et al., 2025a), we report inference-time compute using the *Number of Function Evaluations* (NFE), where each next-token sampling step and each verifier call count as one evaluation. We explain each case below.

**Best-of-$N$ sampling.**   Each of the $N$ sequences requires sampling $T$ tokens and one image-level verification:

$$\text{NFE}_{\text{BoN}} = NT + N.$$

**Beam search.**   With beam size $k$, candidate width $M$, sequence length $T$, and skip length $s$ for sparse verification:

$$\text{NFE}_{\text{beam}} = Tk \; + \; \frac{T}{s}(kM).$$

**Lookahead search.**   Lookahead rolls out each candidate by $L$ steps before verification:

$$\text{NFE}_{\text{lookahead}} = Tk \; + \; \frac{T}{s}(kM)(1 + L),$$

where $L$ is truncated near the end of the sequence.

We note that in practice, the rollout length $L$ may vary across steps (e.g., when rolling out to the end of the sequence), and thus cannot always be treated as a constant multiplier. The above expression is therefore a simplified approximation. Similarly, if the skip length $s$ varies (e.g., under exponentially spaced verification), the formulation should be adjusted accordingly.

**Relation to wall-clock time.** NFE is useful because it is hardware-agnostic and comparable across algorithms, but it does not fully capture realized runtime. In particular, the three functions have different runtimes. We compare wall-clock runtime across search algorithms in Sec. E.4 and Fig. 14, report per-verifier latency in Table 7, and use GFLOPs in Fig. 8 when comparing AR models of different sizes.

# E. Additional Results

## E.1. Other Ordered Tokenization and Generation Schemes

To test whether our findings generalize beyond FlexTok, we evaluate two additional settings: (1) **Semanticist** (Wen et al., 2025a), another 1D ordered tokenizer trained with a similar nested-dropout-based ordering mechanism, and (2) **Infinity** (Han et al., 2025), a scale-wise autoregressive generation framework related to VAR (Tian et al., 2024). These experiments broaden the scope of our study and help disentangle whether the observed gains arise from a specific implementation or from the underlying token ordering structure.

**Results on Semanticist.** To verify that the advantage of ordered token structures is not specific to FlexTok, we evaluate Semanticist (Wen et al., 2025a), a 1D ordered tokenizer that differs from FlexTok in architecture and token space design while sharing a similar nested-dropout-based ordering mechanism. Since the associated AR model is trained for class-to-image generation on ImageNet, we evaluate it on ImageNet-1K (Krizhevsky et al., 2012) with one generated image per class (1K images total), and compare it against the original LlamaGen-L (Sun et al., 2024) model, which uses a 2D grid tokenizer and provides a relatively controlled baseline because Semanticist adopts the same AR backbone on top of its tokenizer.

To better study test-time scaling under this class-conditioned setting, we consider two prompt types for the verifier: (1) **simple prompts** of the form "a photo of a [CLASS_NAME]" and (2) **complex prompts** from ImageNet-1K-VL-Enriched (Visual Layer, 2024), which provide richer caption-level guidance. The latter also offers a useful stress test of whether search can enhance generation when the prior is weak and only provides class-level conditioning. We use CLIPScore (Hessel et al., 2021) as the verifier for all experiments. We apply the **same search algorithms** to both models: Best-of-$N$ sampling ($N=10$), beam search (beam size $= 5$, candidates $= 10$, 4 search steps), and lookahead search (same hyperparameters as beam search, with rollout to the end). Since Semanticist uses 32 tokens while LlamaGen uses 256 tokens, we distribute the 4 search steps proportionally across the sequence: $[1, 4, 16, 32]$ for Semanticist and $[64, 128, 192, 256]$ for LlamaGen.

Table 4 shows that all search algorithms improve over the baseline for both models, but *beam search yields substantially larger gains for the ordered 1D tokenizer*. For example, on simple prompts, beam search improves Semanticist by $+10.42$ CLIPScore points, compared to only $+3.51$ for LlamaGen; on complex prompts, the gap is similarly pronounced ($+12.45$ vs. $+4.04$). These results are consistent with our main findings in the paper and further support the conclusion that the advantage is structural rather than specific to FlexTok. We also observe that complex prompts benefit even more from search than simple prompts, suggesting that *text-guided search can provide meaningful gains even when the underlying autoregressive prior is only class-conditioned*. Example visualizations are provided in Figure 12.

*Table 4.* **Test-time search on Semanticist (1D ordered) vs. LlamaGen-L (2D grid) on ImageNet-1K.** We report CLIPScore (%) under simple and complex prompt guidance. Improvements over the base model are shown in parentheses.

| Model | Tokenizer | Prompt | Base | Best-of-$N$ | Beam Search | Lookahead |
|---|---|---|---|---|---|---|
| Semanticist | 1D ordered | Simple | 74.60 | 81.77 (+7.16) | **85.02 (+10.42)** | 85.51 (+10.91) |
| LlamaGen-L | 2D grid | Simple | 76.85 | 82.50 (+5.64) | 80.36 (+3.51) | 81.96 (+5.11) |
| Semanticist | 1D ordered | Complex | 70.67 | 79.72 (+9.06) | **83.12 (+12.45)** | 83.68 (+13.02) |
| LlamaGen-L | 2D grid | Complex | 73.52 | 80.48 (+6.96) | 77.56 (+4.04) | 79.42 (+5.90) |

**Results on Infinity.** We further evaluate Infinity-2B (Han et al., 2025), a scale-wise autoregressive image generation framework related to VAR (Tian et al., 2024). Unlike standard 2D raster-scan tokenization, Infinity generates images

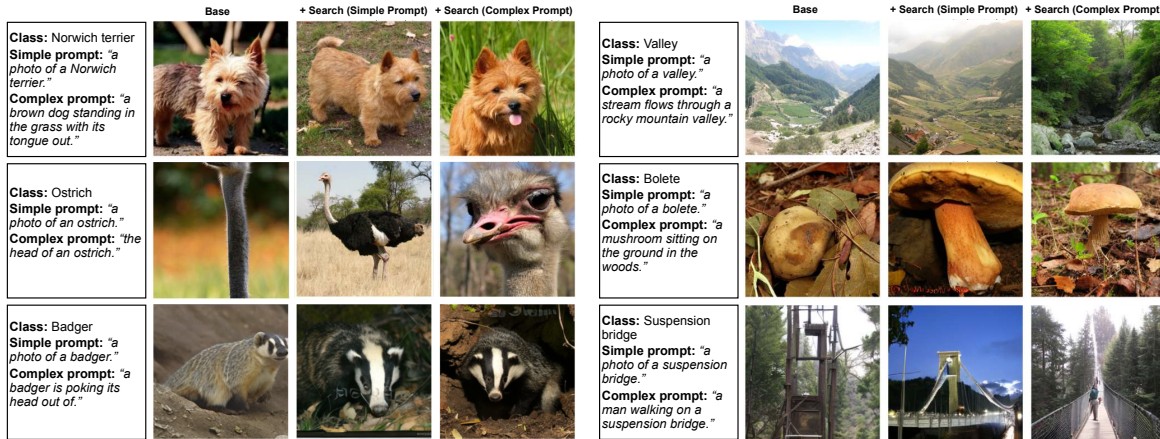

*Figure 12.* **Visualization examples of Semanticist for class-to-image generation on ImageNet.** We compare direct autoregressive generation, beam search with a simple prompt, and beam search with a complex prompt. Beam search generally improves image–text alignment, while complex prompts provide additional guidance beyond class priors. Below each group of images, we show the ImageNet class ID and name, along with the corresponding simple and complex prompts.

*Table 5.* **Comparison of generation paradigms on COCO.** We report CLIPScore (%) for autoregressive decoding and beam search under matched search budgets.

| Model | Janus (Wu et al., 2024a) | Infinity (Han et al., 2025) | FlexTok (Bachmann et al., 2025) |
|---|---|---|---|
| Token structure | 2D grid | 2D multi-scale | 1D ordered |
| AR decoding | 82.3 | 81.9 | 80.4 |
| Beam search | 87.6 (+5.3) | 88.1 (+6.2) | 90.0 (+9.6) |

progressively from low to high spatial resolution, providing a hierarchical ordering over generation steps. This makes it a useful intermediate case between standard 2D grid tokenization and semantically ordered 1D tokenization.

For fair comparison with the other models in our study, we evaluate direct autoregressive decoding and beam search on COCO using CLIPScore. We use beam width $= 5$, candidates $= 10$, and 9 search steps. Since Infinity predicts 13 scales in total and the earliest 4 scales are too coarse to provide reliable verifier guidance, we apply search from step 5 to step 13. This keeps the search budget comparable to our other autoregressive baselines while focusing computation on the stages where intermediate outputs become informative.

Results are shown in Table 5. Infinity benefits substantially from beam search, improving by $+6.2$ CLIPScore points over its autoregressive baseline. This gain is larger than that of Janus ($+5.3$) but smaller than that of FlexTok ($+9.6$). We interpret this as evidence that *ordering itself helps search*, while *semantic coarse-to-fine ordering helps the most*. In particular, Infinity provides a meaningful hierarchy at the spatial-resolution level, which improves searchability relative to standard 2D grid generation, but still appears less effective than a tokenization whose prefixes are explicitly organized by semantic information content.

Together, these additional experiments support the broader conclusion of our paper: the effectiveness of test-time search depends strongly on token structure. Search can improve multiple generation schemes, but the magnitude of the gain varies substantially depending on whether intermediate prefixes expose sufficiently informative structure for the verifier to guide generation effectively.

### E.2. Token Structure Comparison on GenEval

Figure 13 extends the scaling comparison of Figure 6 to the GenEval benchmark, using ImageReward as the verifier. Results are shown for all three search algorithms (Best-of-$N$, beam search, and lookahead search) across both 1D ordered tokens (FlexTok) and 2D grid tokens. The top row reports ImageReward scores and the bottom row reports GenEval compositional accuracy, both as a function of inference compute (NFE). The pattern is consistent with our COCO findings: Beam search yields substantially larger gains for FlexTok than for the 2D grid tokenizer, while Best-of-$N$ shows more similar scaling

across both. The rightmost panel compares each tokenizer under its best-performing algorithm, confirming that FlexTok benefits more from increased inference compute across both metrics. Notably, while FlexTok achieves consistently higher ImageReward scores, the improvement in GenEval accuracy is more modest. This gap likely reflects the mismatch between the continuous verifier signal (ImageReward) and the discrete compositional accuracy metric used by GenEval.

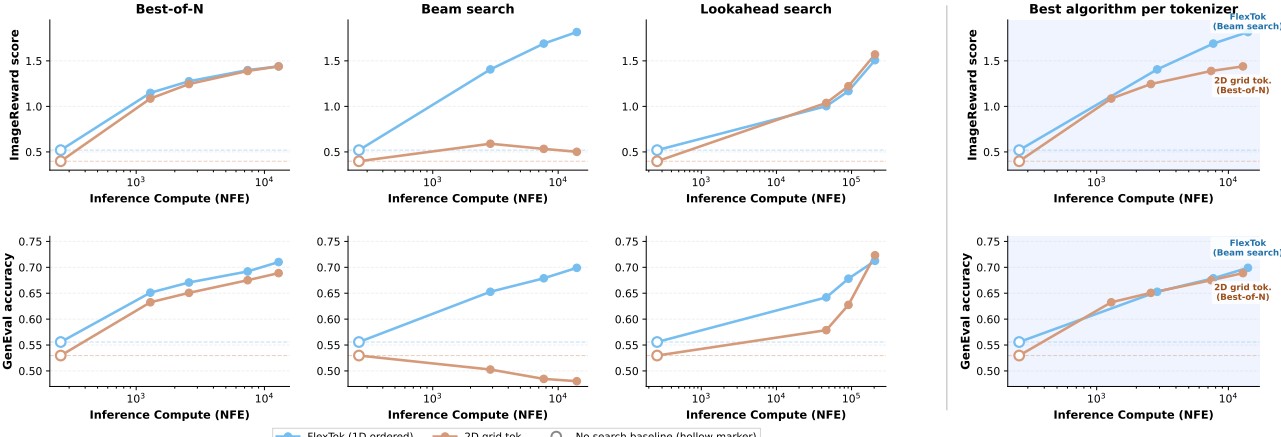

*Figure 13.* **Test-time scaling across token structures on GenEval.** We compare three inference-time search algorithms (Best-of-$N$, beam search, and lookahead search) on two tokenizers: 1D ordered tokens (FlexTok) and a 2D grid tokenizer, evaluated on GenEval using ImageReward as the verifier. Top row: ImageReward score vs. inference compute; bottom row: GenEval accuracy. *The rightmost panel shows each tokenizer paired with its best-performing search algorithm, revealing that FlexTok benefits more from increased inference compute.* Note that while FlexTok achieves higher verifier scores, the improvement in final GenEval accuracy is more modest, likely due to the gap between the verifier signal and discrete task accuracy. NFE (number of function evaluations) measures inference compute; the hollow leftmost marker on each curve denotes the no-search baseline, extended as a color-matched dashed line across each panel for reference.

### E.3. Experimental Scale and Variance Analysis on COCO

We limit our controlled ablations to 300 COCO images due to the substantial computational cost of comprehensive hyperparameter sweeps and lookahead baselines. To study the statistical significance, we further scale key beam search evaluations to a 1,000-image subset of COCO. As shown in Table 6, the variance across 5 random subsets is low, and the performance gap between 1D and 2D tokenizations remains consistent with the 300-image results.

*Table 6.* **Variance analysis on COCO (CLIPScore %).** Results on 300-image and 1K-image subsets. Improvements over the baseline are shown in parentheses. The mean and standard deviation are computed over 5 random subsets for the 300-image setting. Beam search uses a beam width of 5, 10 candidates per step, and 32 search steps.

| Algorithm | 300 Images | 1K Images | 300 Images (Mean $\pm$ Std) |
|---|---|---|---|
| FlexTok Baseline | 80.39 | 80.28 | $80.23 \pm 0.38$ |
| FlexTok Beam Search | 93.44 (+13.05) | 93.72 (+13.44) | $93.53 \pm 0.50$ (+13.30 $\pm$ 0.29) |
| 2D Grid Baseline | 79.06 | 79.19 | $79.25 \pm 0.24$ |
| 2D Grid Beam Search | 81.59 (+2.53) | 81.59 (+2.40) | $81.68 \pm 0.35$ (+2.43 $\pm$ 0.24) |

### E.4. Wall-Clock Runtime Analysis

We report wall-clock runtimes for search algorithms and verifiers to provide a practical reference for practitioners. Figure 14 breaks down wall-clock inference time per algorithm. The dominant cost shifts from AR generation (Best-of-$N$) to detokenization (beam and lookahead search) as search steps increase.

**Verifier runtimes.** Table 7 reports the per-call latency of each verifier on a $256 \times 256$ image using a single GH200 GPU. Lightweight scoring functions such as likelihood and aesthetic score operate in under 25 ms, while ImageReward, HPSv2, and CycleReward are moderately slower (40–60 ms). Rule-based verification via Grounded SAM is the most expensive due

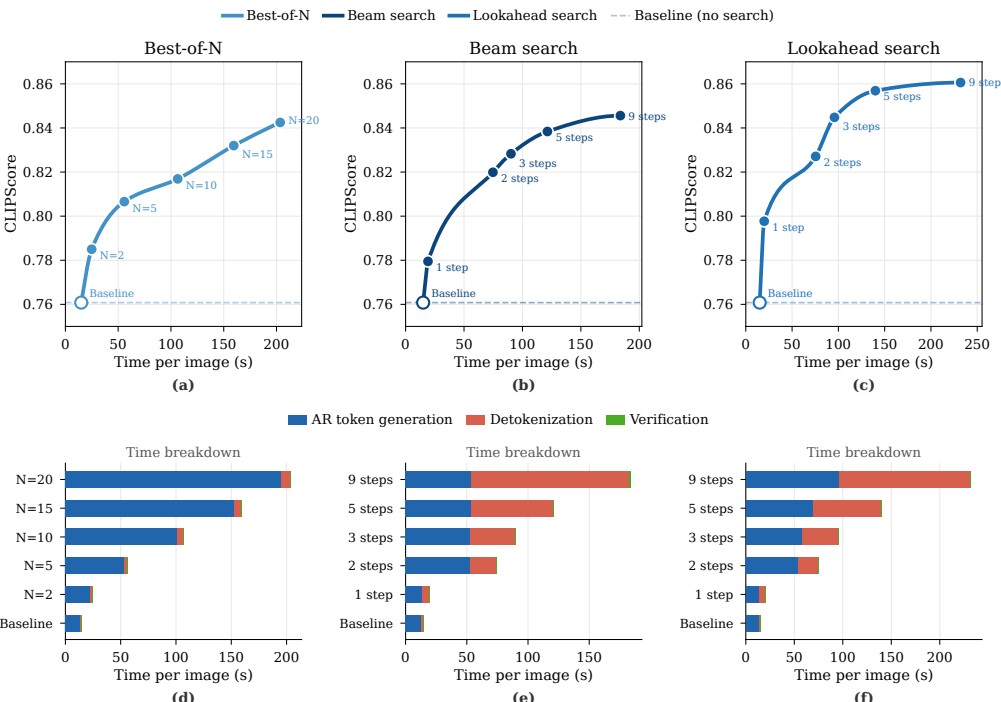

*Figure 14.* **Inference time analysis for different search algorithms (H100 GPU).** Top row (a–c): CLIPScore vs. wall-clock inference time per image for **Best-of-$N$**, **Beam search**, and **Lookahead search** (rollout length $L$=8), respectively. Each point corresponds to one configuration ($N$ or number of search steps), with the open circle marking the no-search AR baseline (dashed line). Bottom row (d–f): empirical wall-clock time breakdown by component for each configuration, showing AR token generation, detokenization, and verification. For Best-of-$N$, inference cost is dominated by repeated AR generation and scales linearly with $N$. For Beam and Lookahead search, detokenization becomes the dominant wall-clock cost as the number of search steps increases, while AR generation time remains roughly constant. This overhead stems from the multi-step nature of the flow-based detokenizer; a faster or single-step detokenizer would directly reduce wall-clock time without affecting the NFE-based scaling behavior shown in Figure 6. Verification cost is negligible across all configurations. All timings are estimated from 20 images on COCO.

to its multi-module pipeline. Across all configurations, verifier cost constitutes a negligible fraction of total inference time compared to AR generation and detokenization, confirming that counting each function evaluation as one NFE unit is a fair and hardware-agnostic measure of inference compute.

*Table 7.* **Verifier runtimes** on a $256 \times 256$ image on GH200 (single GPU). Times are reported in milliseconds (ms).

| Verifier | Time (ms) |
|---|---|
| CLIP | 27.98 |
| Aesthetic | 21.43 |
| ImageReward | 59.16 |
| PickScore | 36.17 |
| HPSv2 | 48.68 |
| Rule-based (Grounded SAM) | 181.31 |
| CycleReward | 46.73 |
| Probability (Likelihood) | 0.06 |

### E.5. Search Hyperparameter Ablations

In this section, we analyze how key search hyperparameters affect performance for both beam search and lookahead search. We focus on the three most influential hyperparameters: (1) beam width, (2) number of search steps (i.e., verification positions), and (3) lookahead length in lookahead search. We leave the candidate width fixed at $M = 10$, since in our

experiments it does not scale as effectively as increasing beam width.[1]

We present results for FlexTok and Janus as representative models. All experiments use a 300-caption COCO subset, with CLIPScore as the primary verifier, and we also report Aesthetic Score and ImageReward for completeness.

**Results on FlexTok.** The results for FlexTok are shown in Table 8. We keep our original default setting ($k$=5, $v$=9, $L$=0) and vary one hyperparameter at a time to study its effect. For **beam width** $k$, we explore $\{2, 5, 10, 15, 20, 25\}$. As $k$ increases, CLIPScore and ImageReward improve, while Aesthetic Score slightly decreases. For the **number of search steps**, we evaluate search with skip lengths $1, 2, 4, \ldots, 128$, which correspond to search and verification counts $256, 128, 64, \ldots, 2$. In our main experiments, we use an exponentially increasing skip schedule consistent with FlexTok training, namely $2^0, 2^1, \ldots, 2^8$, giving $v$=9 search steps. We find that increasing the number of search steps also consistently improves performance, especially for CLIPScore and ImageReward. Lastly, we vary the **lookahead length** from $L$=0 to $L$=256. Lookahead of $L$=32 works best and is comparable to $L$=256, possibly because FlexTok partial sequences with ∼32 tokens already reveal clear semantic structure. Overall, these results show that *our default configuration is a reasonable, lightweight, and efficient setting, while increasing these hyperparameters can further boost performance.* Among them, increasing the number of search steps appears most promising and yields the strongest results in the table. A scaling curve for search steps is shown in Fig. 8 of the main paper.

*Table 8.* **FlexTok hyperparameter sweeps** on a 300-caption COCO subset. Left: beam width. Middle-left: number of search step (v = N/s when uniformly skip length s for token number N). Right: lookahead length. The row corresponding to our **default setting** ($k$=5, $v$=9, $L$=0) is shaded in **gray**. Best values within each block are in **bold**. "Aes" = Aesthetic Score; "IR" = ImageReward.

| Beam Search | | | | | | | | Lookahead Search | | | |
|---|---|---|---|---|---|---|---|---|---|---|---|
| Beam width ($k$), $M$=10, $v$=9 | | | | Number of search steps ($v$), $k$=5, $M$=10 | | | | Lookahead length ($L$), $k$=5, $v$=9, $M$=10 | | | |
| $k$ | CLIP ↑ | Aes ↑ | IR ↑ | $v$ | CLIP ↑ | Aes ↑ | IR ↑ | $L$ | CLIP ↑ | Aes ↑ | IR ↑ |
| 2 | 86.44 | **4.57** | 0.33 | 2 | 81.85 | 4.51 | 0.35 | 0 | 90.04 | 4.49 | 0.33 |
| 5 | 90.04 | 4.49 | 0.33 | 4 | 85.93 | 4.50 | 0.46 | 2 | 91.22 | 4.51 | 0.37 |
| 10 | 92.74 | 4.44 | 0.41 | 8 | 87.68 | 4.52 | 0.49 | 8 | 91.61 | 4.45 | 0.45 |
| 15 | 93.50 | 4.40 | 0.58 | 9 | 90.04 | 4.49 | 0.33 | **32** | **92.84** | **4.47** | **0.44** |
| 20 | 94.63 | 4.39 | 0.60 | 16 | 90.08 | **4.52** | 0.55 | 256 | 92.32 | 4.48 | 0.43 |
| **25** | **101.70** | 4.39 | **0.61** | 32 | 93.45 | 4.52 | 0.56 | | | | |
| | | | | 64 | 98.33 | 4.49 | 0.60 | | | | |
| | | | | 128 | 104.15 | 4.45 | **0.58** | | | | |
| | | | | **256** | **111.48** | 4.41 | 0.43 | | | | |

**Results on Janus.** The results for Janus are shown in Table 9. For beam search, we use the same default setting ($k$=5, $v$=9), and similarly observe that increasing these hyperparameters generally improves performance. For **beam width**, we test $\{2, 5, 10\}$. For the **number of search steps**, we evaluate skip lengths $1, 8, 64, 144$, corresponding to verification counts $576, 72, 9,$ and $4$. In addition, we study **lookahead lengths** of $8, 64, 128,$ and full lookahead. Among these, $L$=128 and full lookahead achieve the best performance. These results highlight that *lookahead is particularly important for 2D grid tokenizers*, whose early tokens provide limited semantic structure.

### E.6. Additional Results on Verifier Analysis

**Per-Category Verifier Breakdown on GenEval** We show per-verifier results in the main paper (Fig. 11). Here in Table 10, we provide the full GenEval category breakdown for all verifiers. Different verifiers excel on different aspects of the benchmark. For example, Grounded SAM achieves the best performance on *Position* and *Color Attribute*, but performs worse on *Single Object*, *Colors*, and *Counting*. Likelihood improves most categories except *Position*. ImageReward and the ensemble perform similarly strong overall, with the ensemble achieving the best overall accuracy among learned verifiers. The official GenEval evaluator serves as an upper bound.

**Verifier Comparison on COCO** Following the GenEval analysis, we also evaluate FlexTok with beam search on the COCO 300-caption subset using different verifiers, including leave-one-out and verifier ensemble settings. The full results

---

[1]Intuitively, increasing $M$ expands the search but retains the same number of prefixes, whereas increasing beam width expands and preserves more candidates simultaneously, yielding better returns.

*Table 9.* **Janus hyperparameter sweeps** on a 300-caption COCO subset. Left: beam width. Middle-left: Number of search steps. Right: lookahead length. The **default setting** ($k$=5, $v$=9, $L$='All') is shaded in **gray**. Best values within each block are in **bold**. "Aes" = Aesthetic Score; "IR" = ImageReward.

| Beam Search | | | | | | | | Lookahead Search | | | |
|---|---|---|---|---|---|---|---|---|---|---|---|
| Beam width ($k$), $M$=10, $v$=9 | | | | Number of search steps. ($v$), $k$=5, $M$=10 | | | | Lookahead length ($L$), $k$=5, $v$=9, $M$=10 | | | |
| $k$ | CLIP ↑ | Aes ↑ | IR ↑ | $v$ | CLIP ↑ | Aes ↑ | IR ↑ | $L$ | CLIP ↑ | Aes ↑ | IR ↑ |
| 2 | 85.37 | **4.83** | 0.21 | 4 | 87.49 | 4.80 | **0.24** | 0 | 87.59 | 4.80 | 0.15 |
| 5 | 87.59 | 4.80 | 0.15 | 9 | 87.59 | 4.80 | 0.15 | 8 | 88.21 | 4.81 | 0.29 |
| 10 | **89.02** | 4.79 | **0.27** | 72 | 88.19 | **4.81** | 0.17 | 64 | 90.58 | 4.83 | 0.33 |
| | | | | 576 | **92.18** | 4.61 | -0.19 | 128 | 91.49 | 4.83 | 0.26 |
| | | | | | | | | **All** | **92.27** | **4.84** | **0.38** |

*Table 10.* **Verifier performance on GenEval categories.** Numbers are integers. The **Overall** score is normalized to 0–100. Best values in each column (excluding the oracle GenEval row) are highlighted in **bold**. The official GenEval evaluator is shown in gray as an oracle upper bound.

| Method | Single Obj. | Position | Two Obj. | Colors | Color Attr. | Counting | Overall ↑ |
|---|---|---|---|---|---|---|---|
| FlexTok Base | 95 | 16 | 56 | 80 | 35 | 59 | 57 |
| Likelihood | **100** | 14 | 66 | 84 | 36 | 63 | 60 |
| CLIPScore | 96 | 20 | 72 | 87 | 41 | 63 | 63 |
| AestheticScore | 99 | 13 | 68 | 81 | 42 | 50 | 59 |
| CycleReward | 98 | 18 | 69 | 86 | 42 | 59 | 62 |
| HPSv2 | **100** | 16 | 77 | 89 | 42 | **74** | 66 |
| ImageReward | 98 | 27 | **81** | 87 | 41 | 71 | **67** |
| Grounded SAM | 94 | **31** | 51 | 85 | 44 | 58 | 60 |
| PickScore | 98 | 12 | 69 | 82 | 37 | 70 | 61 |
| Ensemble | **100** | 17 | 74 | **89** | **47** | **74** | **67** |
| GenEval (Oracle) | 100 | 41 | 83 | 91 | 60 | 81 | 76 |

are shown in Figure 15. We observe a similar trend as in GenEval: different verifiers specialize in different aspects, but the *ensemble* consistently achieves the best average rank and is almost always the second-best method for each individual metric. This further confirms that combining complementary verifier signals yields stronger and more stable performance.

**Verifier Score Dynamics During Search**   To better understand how different verifiers interact during optimization, we analyze how the scores of all verifiers evolve as search progresses when *one* verifier is used as the optimization target. Figure 16 visualizes these trajectories.

We observe that when optimizing most verifiers, not only does the target verifier score increase steadily, but many other verifier scores also improve. This suggests that the majority of verifiers we use capture a broad notion of visual quality or semantic alignment that tends to correlate across metrics. Notably, optimizing ImageReward raises Aesthetic Score more strongly than optimizing CLIPScore or Grounded SAM, indicating that ImageReward encourages perceptual and stylistic improvements, whereas the other two verifiers primarily drive semantic alignment. In addition, optimizing the rule-based Grounded SAM verifier also yields consistent gains across other verifier dimensions. This implies that strengthening spatial grounding often contributes to improved global alignment. Moreover, because the rule-based verifier saturates at a score of 1 once all spatial constraints are satisfied, it is less susceptible to over-optimization or verifier hacking, reducing the likelihood of producing degenerate solutions.

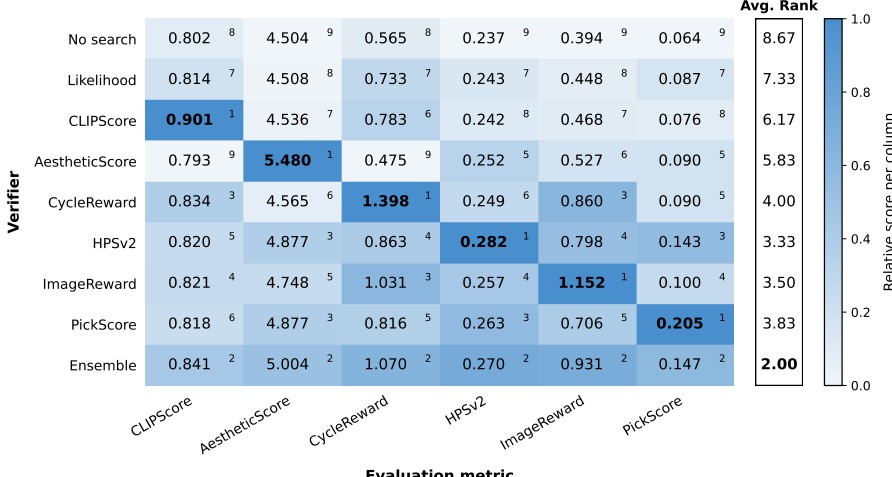

*Figure 15.* **Comparison of different verifiers on COCO.** Each row reports search using one verifier. All methods use the same beam search algorithm on FlexTok. The best score in each column is highlighted in bold. The superscript in each cell represents the rank within that column's metric, and the last column reports the average of these column-wise ranks, providing an overall rank for each verifier.

## F. Additional Visualizations

### F.1. Visualization for Different AR Priors

We compare three prior settings: a conditional AR prior (the standard text-conditioned FlexTok model), an unconditional AR prior (the same model without text conditioning), and a uniform prior. Results for different priors during search are presented in Figures 17–20.

### F.2. Visualization for Different Verifiers

To better illustrate how different verifiers influence the search outcome, we compare the images produced by FlexTok using direct autoregressive decoding and beam search guided by various verifiers. Figure 21–25 show examples from the GenEval benchmark using verifiers including ImageReward, CLIPScore, Grounded SAM, Aesthetic Score, CycleReward, HPSv2, and the verifier ensemble. Different verifiers exhibit distinct preferences; for example, Aesthetic Score often favors better image quality and aesthetics, while CLIPScore and ImageReward tend to better preserve object semantics and counting. The ensemble generally provides the most balanced behavior.

### F.3. Visualization for Zero-shot Multimodal Control

We provide additional qualitative results on the DreamBench++ benchmark in Figures 26–28. We first compare direct AR generation against DreamSim-guided search on several additional subjects, then show a larger set of search-only qualitative examples. As in the main paper, images are generated using FlexTok, and DreamSim is used as the verifier for search. Each example consists of a reference identity image followed by multiple generated images conditioned on different prompts.

## G. Failure Case Analysis

We discuss representative failure cases that happen in test-time search: (1) verifier hacking and (2) prior bottleneck.

**Verifier hacking.**    A fundamental limitation of test-time search is its reliance on the robustness of the external verifier. When the search budget becomes large (e.g., high beam width or many search steps), the optimization process may overfit to the verifier and exploit its blind spots. In practice, this can lead to visually implausible or semantically inconsistent images that nevertheless achieve high verifier scores. For example, in Figure 16, when optimizing CLIP or Grounded SAM scores, the optimized score continues to increase, while the aesthetic score may decrease. Similarly, when optimizing only for aesthetic score, task performance (e.g., GenEval accuracy) may drop. Similar trade-offs can be observed in Figure 21–25 when optimizing different verifiers.

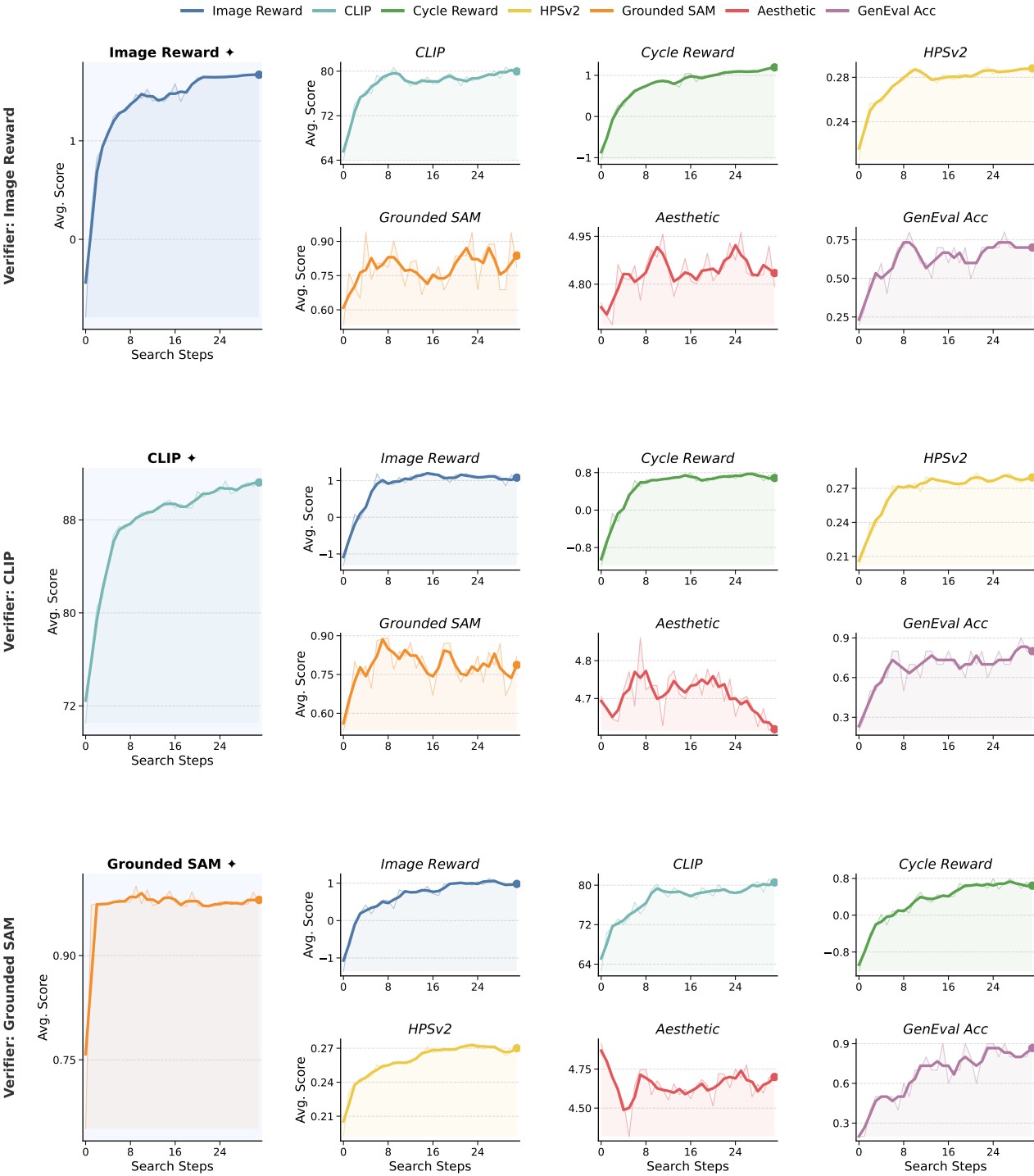

*Figure 16.* **Verifier score trajectories during search.** Each panel shows how optimizing one verifier affects all other verifier signals as well as GenEval accuracy. Curves are averaged over 15 prompts from GenEval using FlexTok with beam search on the first 32 tokens. Note that we only show verifier scores where they are comparable during the search process; we exclude likelihood because it always increases with longer token sequences, and also PickScore because it is affected by how similar other images are.

**Prior bottleneck.**   While 1D ordered tokens help establish global semantics early in the generation process, test-time search cannot recover information that is missing or poorly modeled by the autoregressive prior. In such cases, search may refine local details but fail to correct global structural errors. For example, in Figure 10, under a uniform prior setting, the searched results fail to generate key semantic elements (e.g., "wine"), due to the lack of appropriate object priors in the initial generation. In Figure 9, even after search, the generated images still deviate from the reference image (e.g., in the "fog" case). Although search improves over AR decoding, the results remain limited by the weak or misaligned prior.

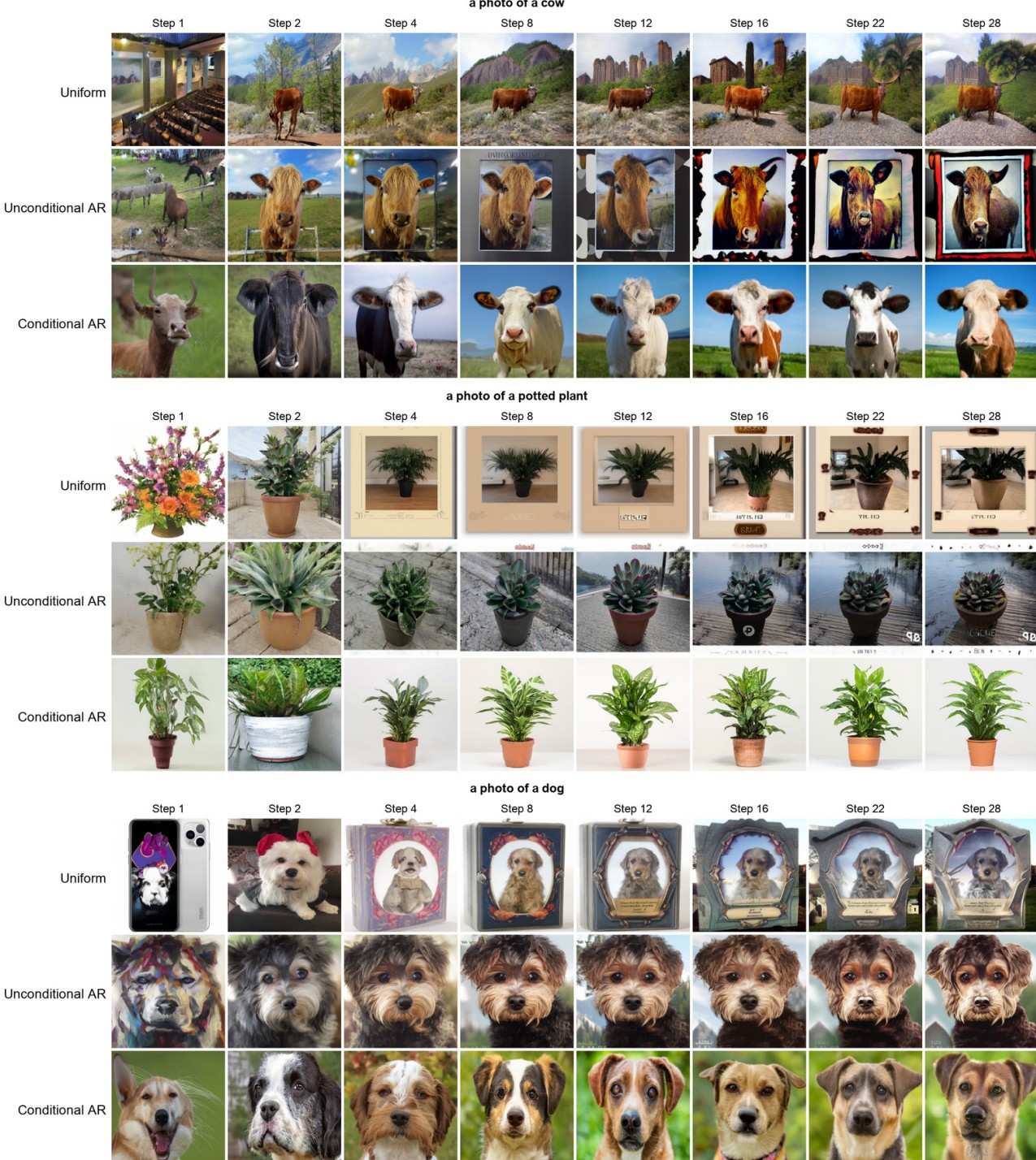

*Figure 17.* **Visual comparison when searching with different AR priors (Examples 1–3).** Beam search guided by different AR priors on the GenEval benchmark.

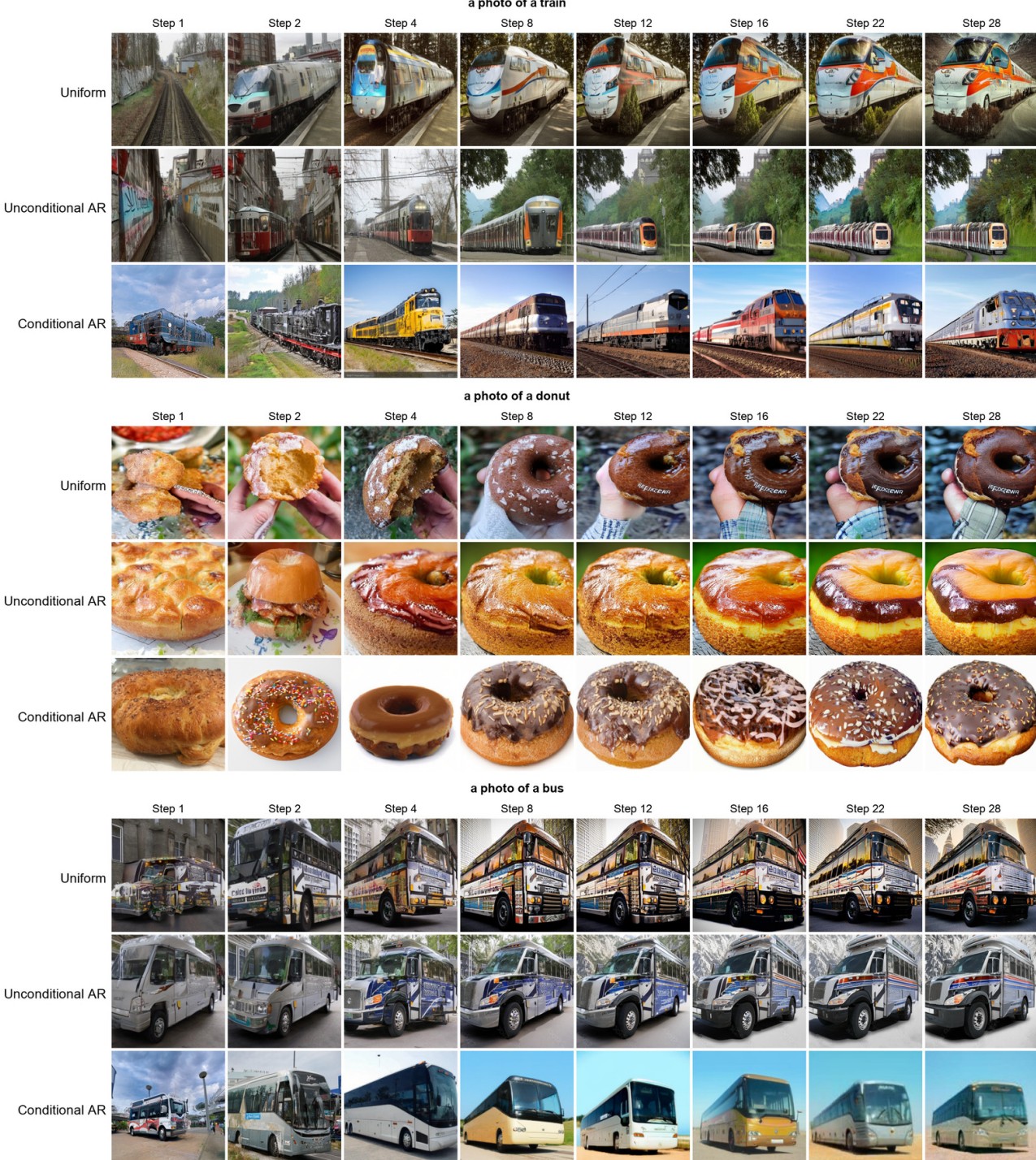

*Figure 18.* **Visual comparison when searching with different AR priors (Examples 4–6).** Beam search guided by different AR priors on the GenEval benchmark.

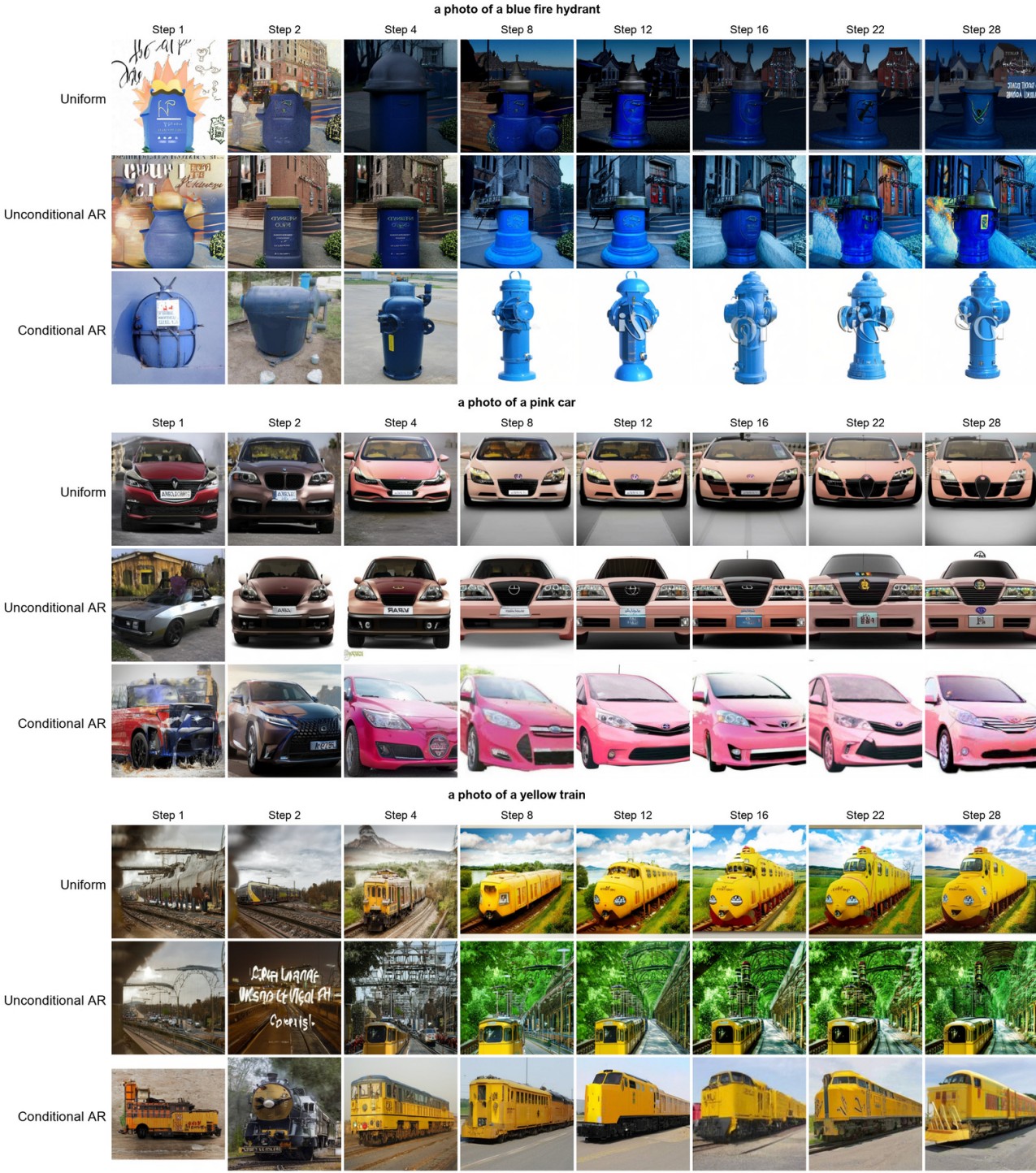

*Figure 19.* **Visual comparison when searching with different AR priors (Examples 7–9).** Beam search guided by different AR priors on the GenEval benchmark.

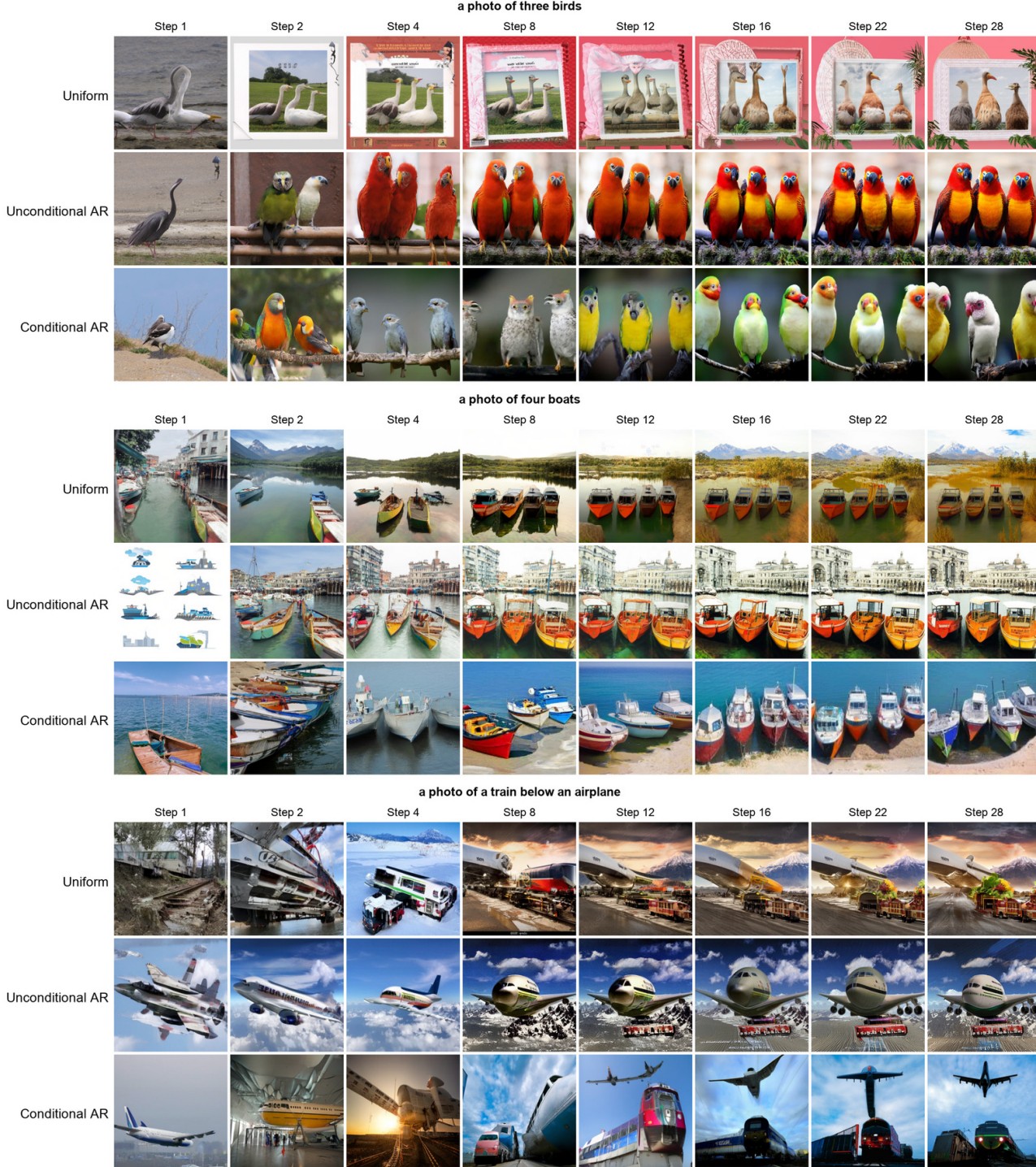

*Figure 20.* **Visual comparison when searching with different AR priors (Examples 10–12).** Beam search guided by different AR priors on the GenEval benchmark.

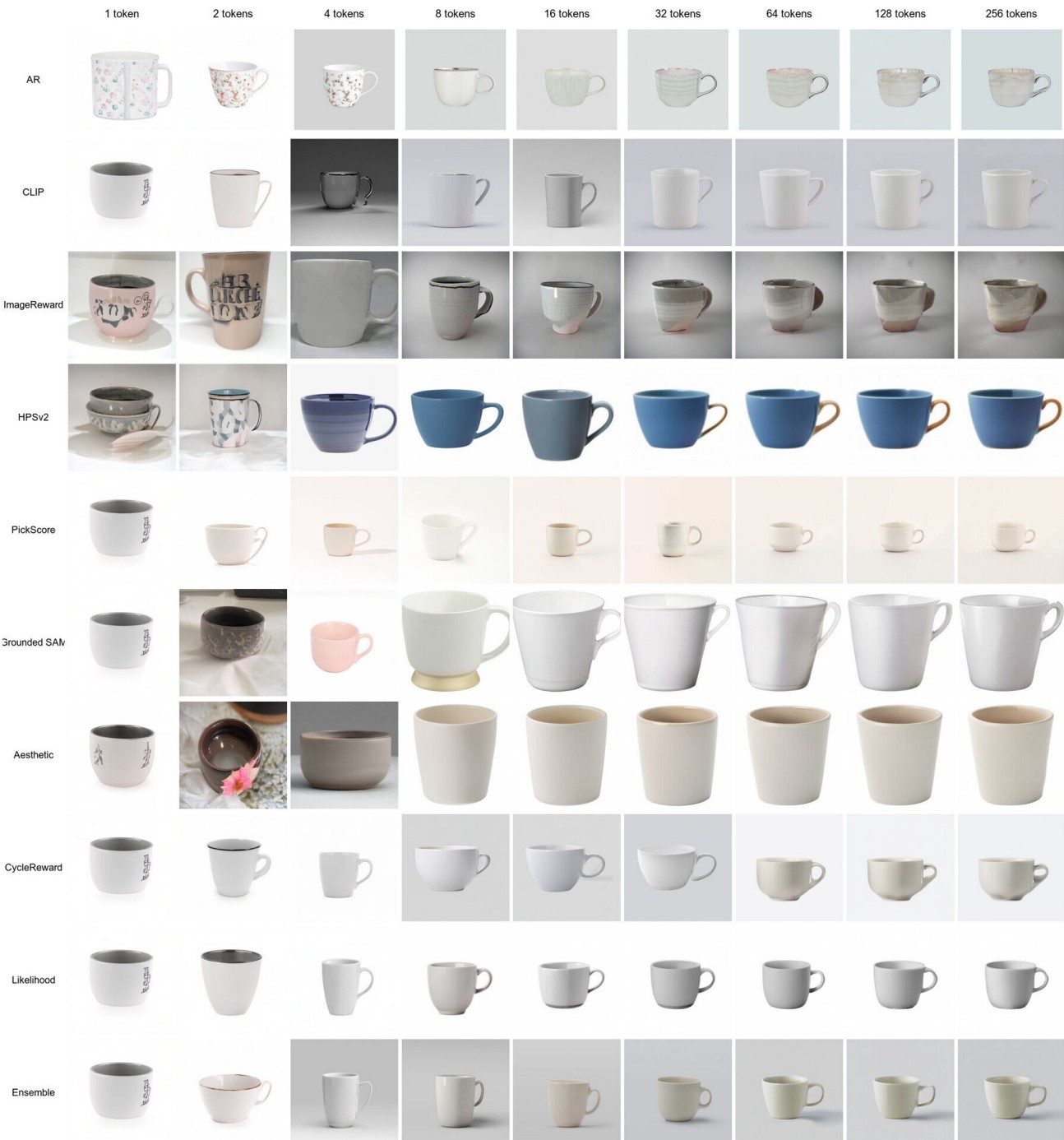

*Figure 21.* **Generation trajectories during verifier-guided search up to 256 tokens: cup.** We show intermediate outputs for the prompt "a photo of a cup" at token positions 1, 2, 4, 8, 16, 32, 64, 128, 256. Even for a simple single-object prompt, different verifiers induce noticeably different search paths in object shape, texture, and realism before converging.

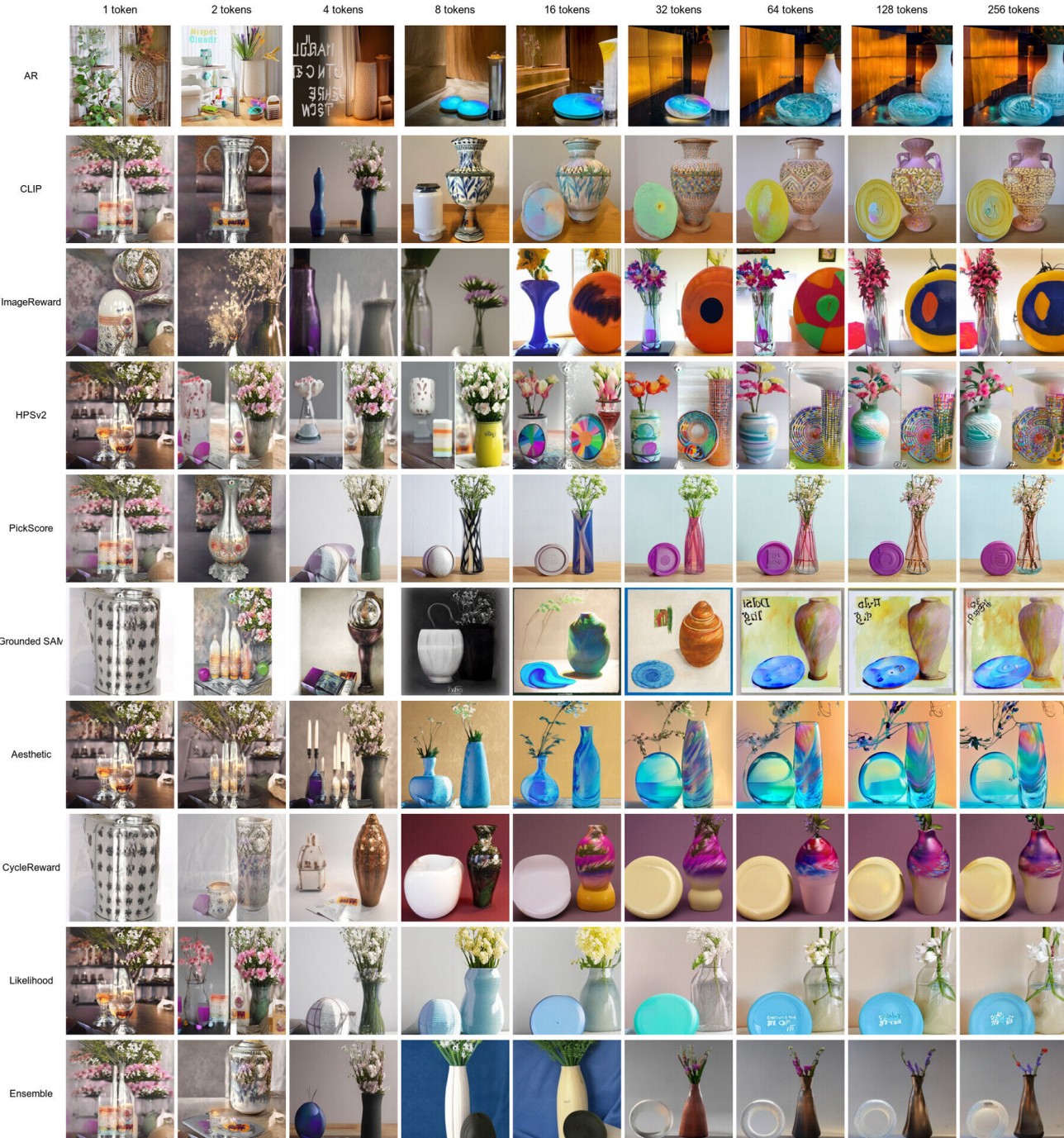

*Figure 22.* **Generation trajectories during verifier-guided search up to 256 tokens: frisbee and vase.** This prompt highlights how different verifiers handle a two-object composition with competing semantics. Some verifiers lock onto one object earlier, while others preserve both objects more reliably over the full search trajectory.

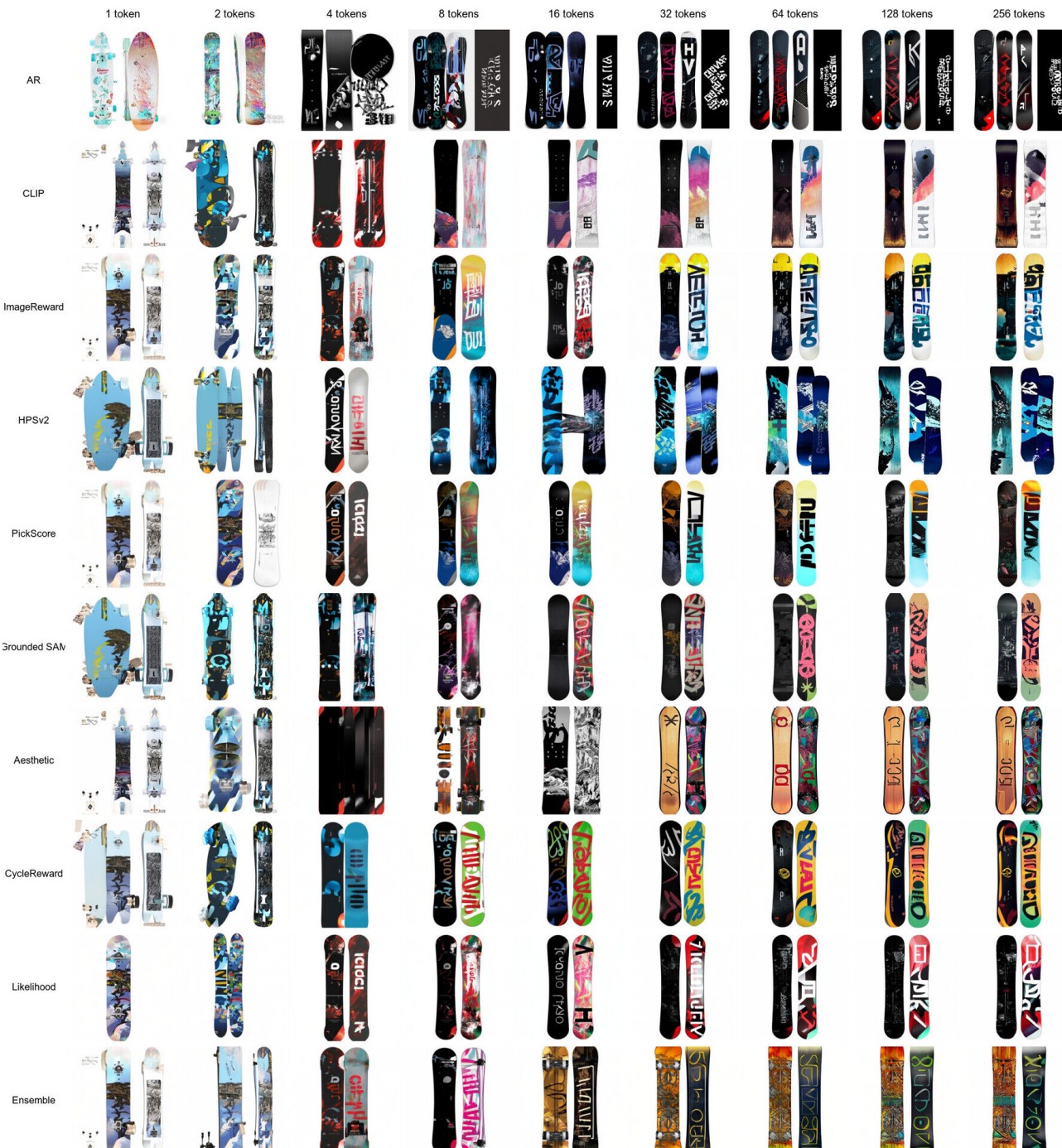

*Figure 23.* **Generation trajectories during verifier-guided search up to 256 tokens: two snowboards.** This counting-and-category prompt shows how verifiers differ in how quickly they commit to the correct duplicated object structure. Some prioritize realistic texture early, while others more directly organize the scene around the requested count.

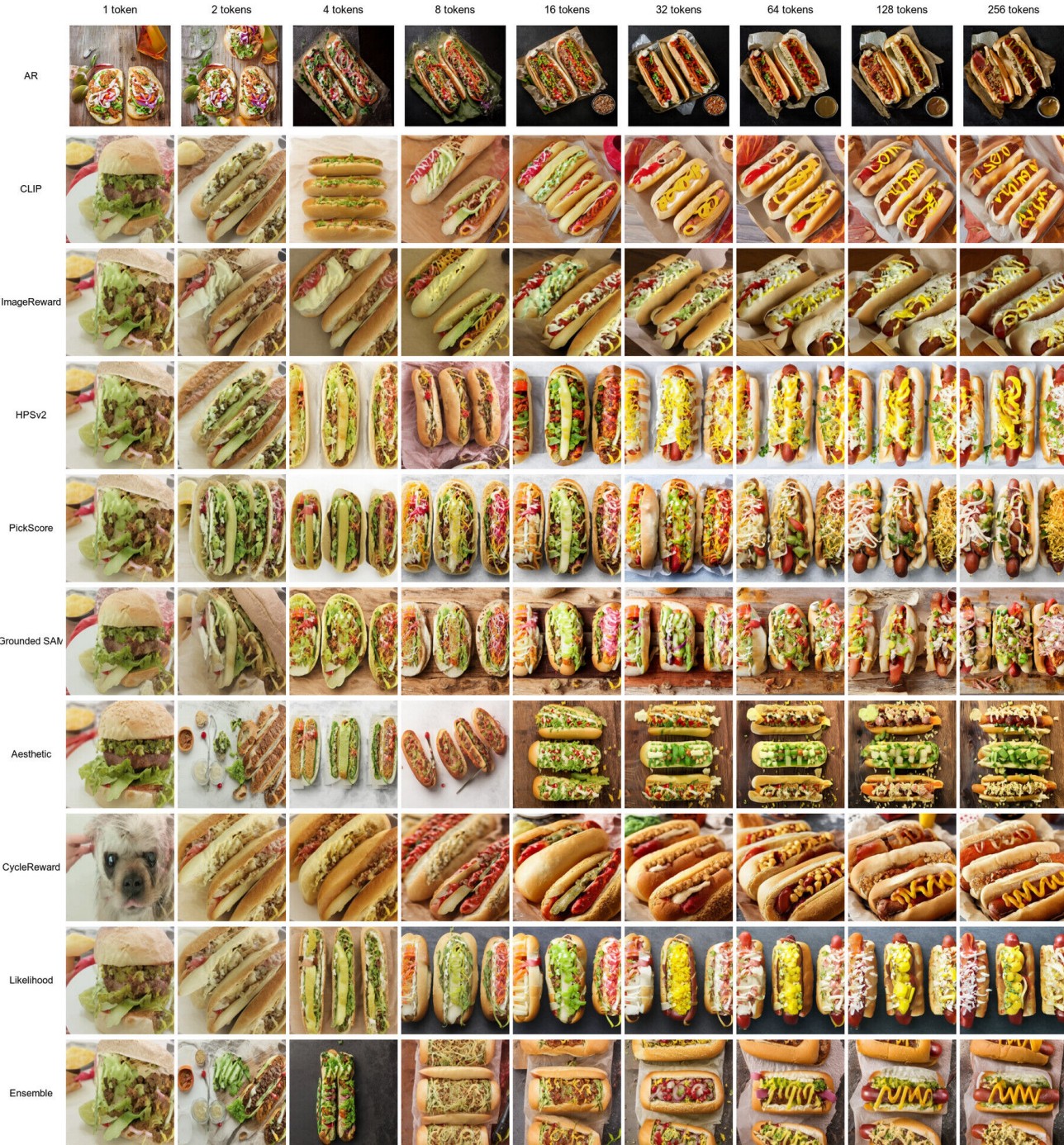

*Figure 24.* **Generation trajectories during verifier-guided search up to 256 tokens: three hot dogs.** This counting prompt illustrates how verifier choice changes the search path even when the target concept is simple. Alignment-focused verifiers improve object identity quickly, while the ensemble and structural verifiers more reliably organize the scene toward the requested count.

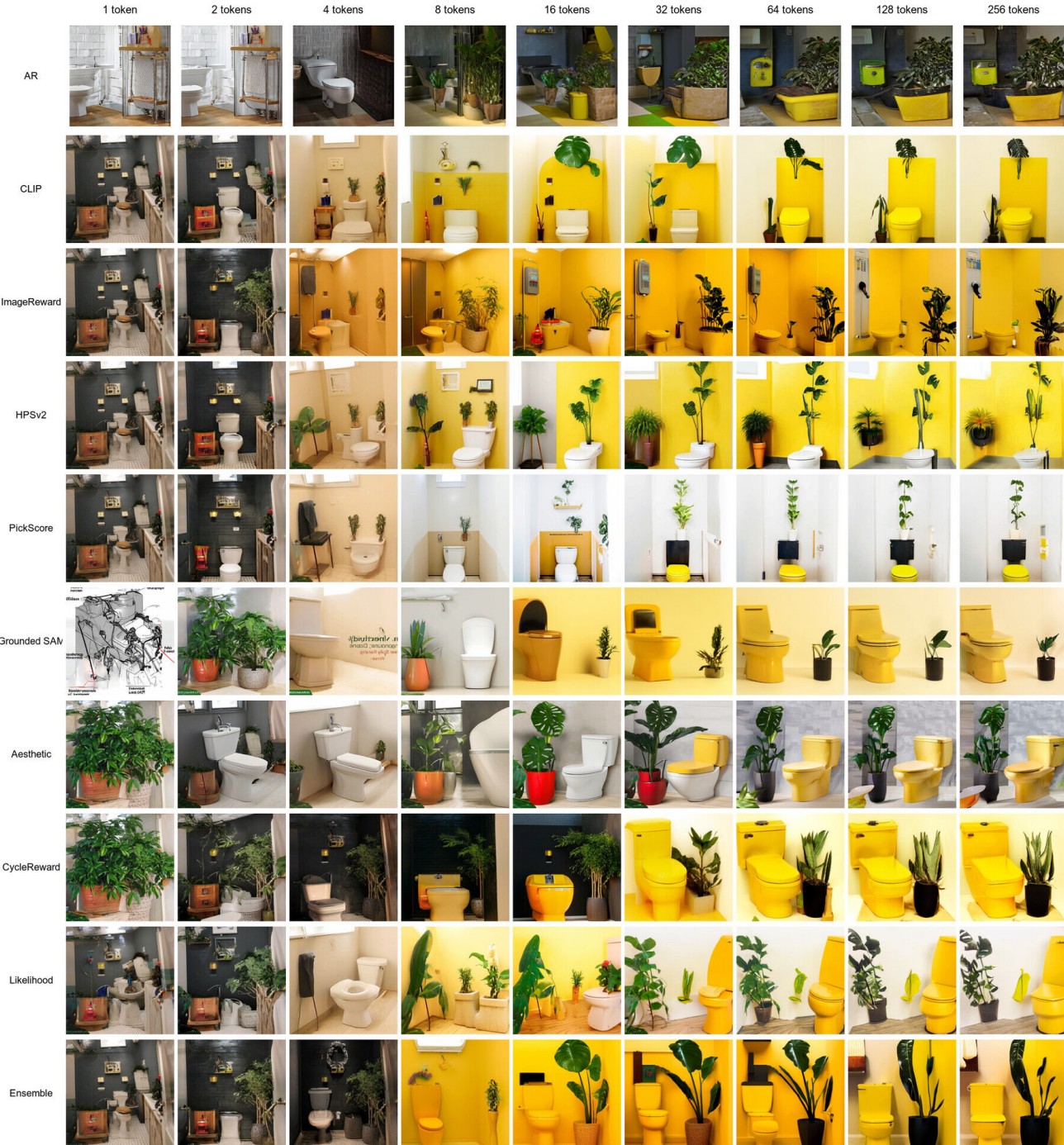

*Figure 25.* **Generation trajectories during verifier-guided search up to 256 tokens: black potted plant and yellow toilet.** This prompt emphasizes unusual object and color combinations. Different verifiers stabilize realism and layout at different rates; structural and ensemble guidance more reliably steer the search toward the requested plant–toilet composition over the full trajectory.

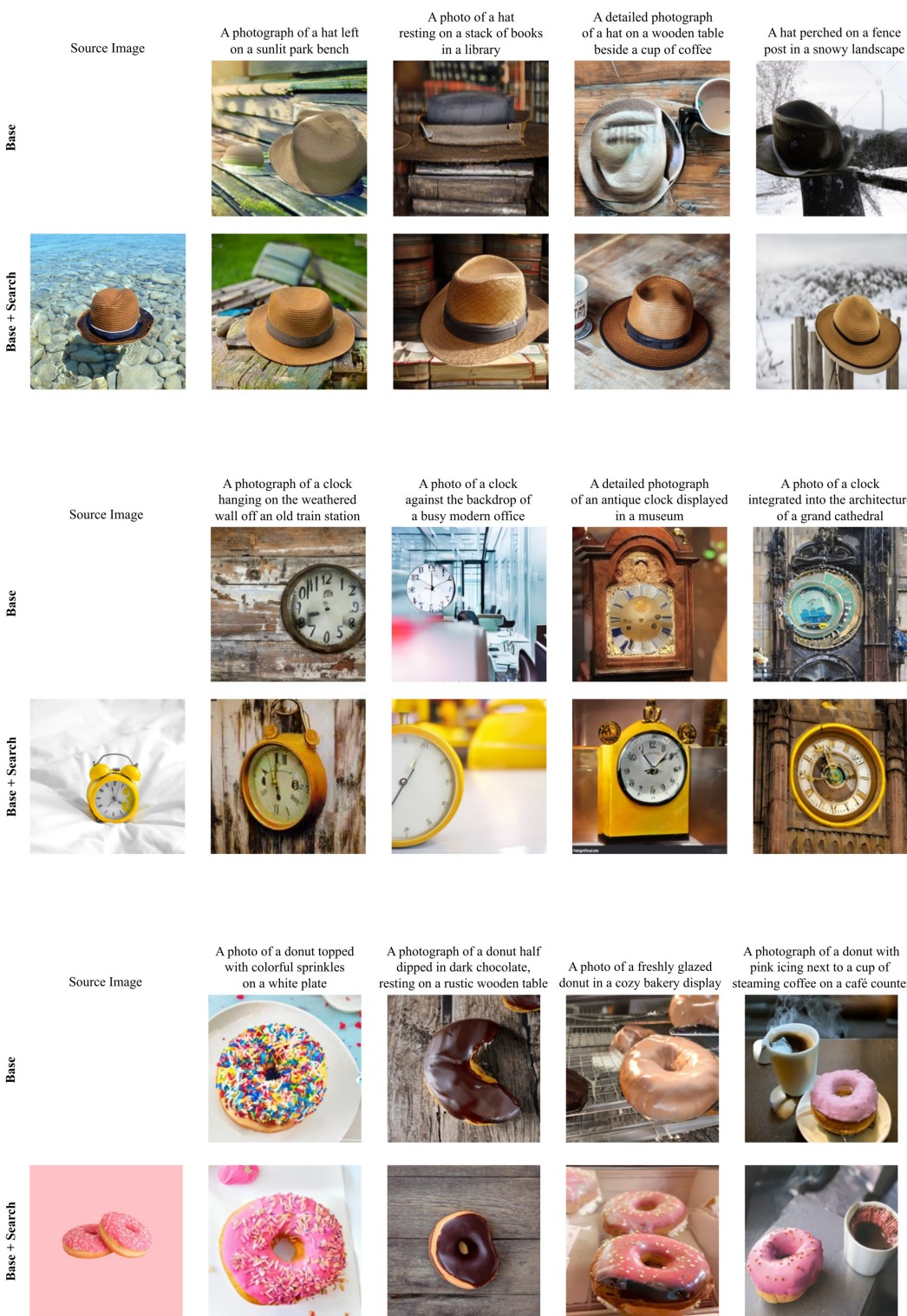

*Figure 26.* **DreamBench++ comparison between direct AR generation and DreamSim-guided search (Examples 1–3).** Each panel compares the direct AR baseline (*Base*, top row) against verifier-guided search (*Base + Search*, bottom row) for the same reference subject and prompt set. Search consistently improves identity preservation and prompt-conditioned scene adaptation.

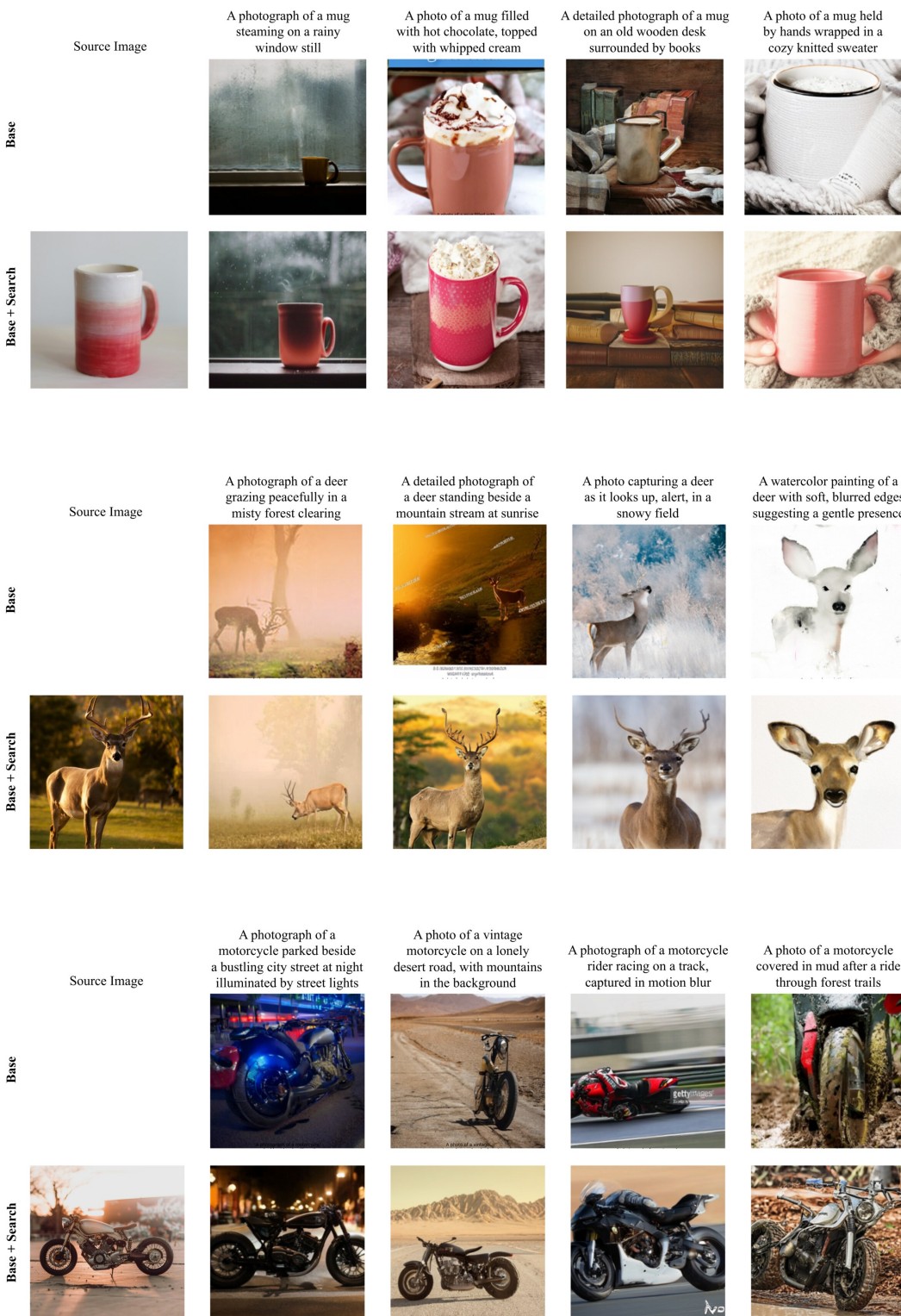

*Figure 27.* **DreamBench++ comparison between direct AR generation and DreamSim-guided search (Examples 4–6).** Additional subjects using the same visualization format as Figure 26. The bottom row in each panel shows that search better preserves subject identity while adapting to diverse prompt contexts.

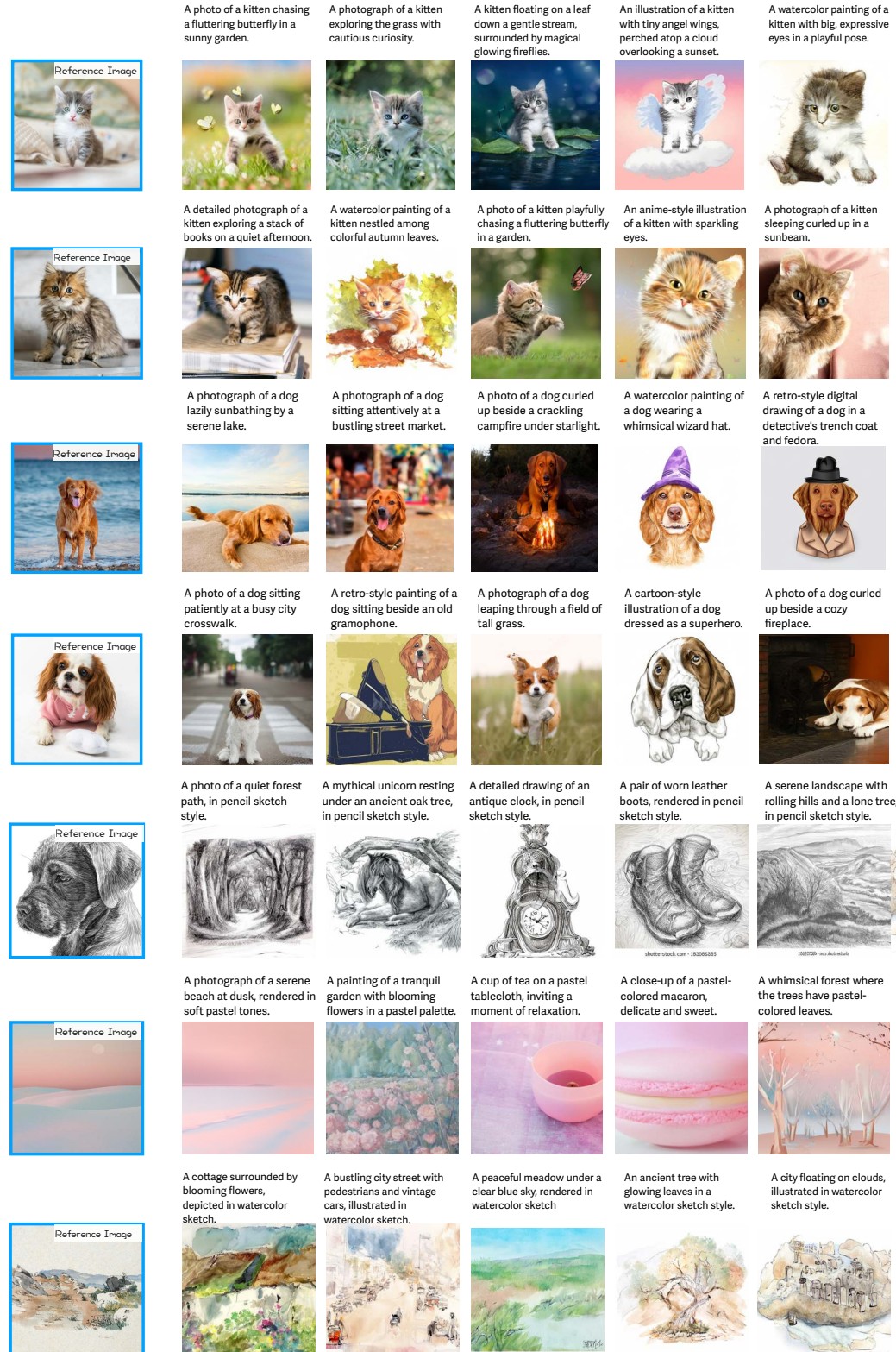

*Figure 28.* **DreamBench++ reference images and generated results.** Each row corresponds to a single subject from Dream-Bench++ (Peng et al., 2024), with the leftmost column showing the reference image and the remaining columns showing images generated by FlexTok (Bachmann et al., 2025) using beam search guided by the DreamSim verifier (Fu et al., 2023).

