# OpenReview forum: "(1D) Ordered Tokens Enable Efficient Test-Time Search"
_ICML.cc/2026/Conference — ICML 2026 regular_

### Official Review · Reviewer_BUbr · 2026-02-21

**Soundness:** 3
**Presentation:** 3
**Significance:** 2
**Originality:** 2
**Overall Recommendation:** 4
**Confidence:** 3

**Summary:**

This paper studies how tokenization structure influences the effectiveness of test-time search in autoregressive image generation. It compares conventional 2D grid tokens with 1D coarse-to-fine ordered tokens and hypothesizes that ordered tokens produce semantically meaningful intermediate states that better support search with external verifiers. The authors propose a Search-over-Tokens (SoT) framework combining autoregressive priors, various search strategies, and verifier models. Experiments show that models using ordered tokens achieve stronger test-time scaling than grid-based models and enable training-free text-to-image generation and zero-shot multimodal control.

**Compliance With Llm Reviewing Policy:**

Affirmed.

**Final Justification:**

The rebuttal has addressed most of my concerns, and the perspective explored is quite novel.

**Key Questions For Authors:**

See the weakness. If most questions are resolved, I will increase my score.

**Limitations:**

yes

**Strengths And Weaknesses:**

Strengths
1. *Novel perspective on tokenization*. An important concept assessed by this paper is the relationship between token ordering and searchability. Rather than proposing a new model architecture, the paper highlights token structure as a key bottleneck or enabler for inference-time improvement, which is a fresh angle compared to most work focusing on training methods.

2. *Strong empirical analysis.* The paper provides extensive experiments Controlled comparisons between 1D ordered and 2D grid tokenizers, Multiple search algorithms, Diverse verifiers, Scaling across model sizes, Zero-shot multimodal control, Ablations on autoregressive priors.

Weaknesses
1. *Limited conceptual novelty beyond empirical observation.* While the empirical findings are interesting, the paper does not introduce fundamentally new algorithms or theoretical insights. The main contribution is largely an empirical demonstration that existing ordered tokenizers work better with search.

2. *Dependence on a specific tokenizer.* Most experiments rely heavily on FlexTok. It is unclear whether the conclusions generalize to other ordered tokenization schemes or modalities.

3. *Lack of theoretical analysis.* The paper provides intuitive explanations (e.g., semantic intermediate states) but lacks formal analysis of why ordered tokens enable better search or scaling.

---

> ### Author Rebuttal · Authors · 2026-03-31
>
> We sincerely thank the reviewer for the thoughtful and constructive feedback. We are aligned with the core concerns regarding theoretical grounding and generalization, and clarify both below.
>
> # Theoretical analysis (W1 & W3)
> We agree the current version lacks a clear formalization of why ordered tokens improve search, and are happy to clarify. Below, we provide a concise theoretical analysis connecting the error bound of heuristic search to token ordering, which will be made explicit in the revision.
>
> *Key idea.* If image tokens are organized with an ordering that better facilitates verifier readout from prefix tokens, search becomes more efficient than with unordered or poorly ordered tokenizations, analogous to KD-trees and PCA-trees [d2]. We formalize this by relating the search gap to the reconstruction ambiguity of intermediate states.
>
> *Setup.*  Let $x^* = \arg\max_x g(x, c)$ be the optimal image for a given verifier $g$ and context $c$, and let $\hat{x}$ denote the image found by a heuristic search algorithm.  Define the search gap $\Delta = g(x^*,c) - g(\hat{x},c)$. Let $x_{1:t}$ denote the decoded intermediate image obtained from a token prefix via a partial decoder, and define the optimal continuation $F_t(x_{1:t}) = \max_{x_{t+1:T}} g(x_{1:T},c)$. The heuristic error is $B_t = \sup_{x_{1:t}} |F_t(x_{1:t}) - g(x_{1:t},c)|$. Let $t_0$ be the step at which the optimal prefix $x^{1:t}$ is pruned, and define the continuation suboptimality $\eta_{t_0} = F_{t_0}(\hat{x}_{1:t_0}) - g(\hat{x},c)$.
>
> **Proposition 1.** The search gap satisfies: $\Delta \le 2B_{t_0} + \eta_{t_0}$.
>
> *Proof sketch.* Assume the algorithm diverges at step $t_0$ (i.e., $g(\hat{x}\_{1:t\_0},c) \ge g(x^\*\_{1:t\_0},c)$), then $g(x^\*,c) = F\_{t\_0}(x^\*\_{1:t\_0}) \le g(x^\*\_{1:t\_0},c) + B\_{t\_0} \le g(\hat{x}\_{1:t\_0},c) + B\_{t\_0} \le F\_{t\_0}(\hat{x}\_{1:t\_0}) + 2B\_{t\_0}$. Substituting $F\_{t\_0}(\hat{x}\_{1:t\_0}) = g(\hat{x},c) + \eta\_{t\_0}$ yields the bound. *(The AR prior can be incorporated by replacing $g(x,c)$ with $g(x,c) + \lambda \log p\_\omega(x)$.)*
>
> **Proposition 2.** If $g$ is $L$-Lipschitz and $\sup\_{x \in \mathcal{C}(x\_{1:t})} \|x\_{1:t} - x\|\_2 \leq \epsilon\_t$, then $B\_t \leq L\epsilon\_t$.
>
> *Proof sketch.* For any completion $x$, $|g(x\_{1:t}) - g(x)| \leq L \|x\_{1:t} - x\|\_2$. Taking supremum over completions and prefixes gives $B\_t \leq L\epsilon\_t$.
>
> Combining the two propositions, we obtain $\Delta \le 2B_{t_0} + \eta_{t_0}$ and $B_t \le L\epsilon_t$, yielding $\Delta \le 2L\epsilon_{t_0} + \eta_{t_0}$.
>
> *Implication.* (a) *1D ordered tokenizers* explicitly minimize intermediate reconstruction error via nested dropout ($\mathcal{L}\_{\text{nested}} = \mathbb{E}\_t[\|x\_{1:t} - x\|\_2^2]$), keeping $\epsilon\_t$ small throughout generation. As global structure is captured early, $\epsilon\_t$ (and hence $\Delta$) decreases rapidly with $t$, providing reliable early-stage guidance. (b) *2D grid tokenizers* typically only constrain $\epsilon\_t$ at $t=T$; at intermediate steps ($t \ll T$), large regions remain unconstrained, so $\epsilon\_t$ (and thus $\Delta$) can remain large until critical features are generated.
>
> The formulation is consistent with our empirical observation that 1D ordered tokens facilitate better test-time scaling, particularly when guided by a verifier where early prefixes trained with nested dropout already provide reasonable score estimates.
>
> # Other ordered tokenization schemes  (W2)
>
> We agree that evaluating additional ordered tokenization schemes broadens the scope of our claims. To show that, we conducted experiments with Semanticist (Wen et al., 2025), which enforces similar coarse-to-fine ordering via nested dropout but differs in architecture and token space. As shown, Semanticist shows stronger test-time scaling than LlamaGen (a comparable 2D grid tokenizer), especially under beam search: +10.42 vs. +3.51 of CLIPScore (%) on simple prompts, and +12.45 vs. +4.04 on complex prompts (full results in Table A1, response to Reviewer FHzW).
>
> We also evaluated Infinity (Table A2), a scale-wise autoregressive paradigm related to VAR. Under matched search budgets, Infinity benefits from beam search (+6.2 CLIPScore), outperforming Janus (+5.3) but below FlexTok (+9.6). This provides a consistent picture: ordering is beneficial for search, and semantic coarse-to-fine ordering appears more effective (see Table A2 in our response to Reviewer FHzW).
>
> Overall, these results support our main claim that the structure of ordered tokens plays an important role in enabling effective search, rather than the effect being tied to a specific tokenizer. Extending this tokenization–search perspective to other modalities is a natural direction for future work.
>
> [d1] Rippel, O., & Adams, R. P. *Learning Ordered Representations with Nested Dropout.* ICML, 2014.
> [d2] McNames, J. *A Fast Nearest-Neighbor Algorithm Based on a Principal Axis Search Tree.* IEEE TPAMI, 2021.

---

> > ### Author Rebuttal · Reviewer_BUbr · 2026-04-03
> >
> > The authors have addressed my concern, so I will keep my score.

---

> > > ### Author Response · Authors · 2026-04-08
> > >
> > > We thank the reviewer for the acknowledgement and for noting that the concerns have been addressed. We respectfully note that option (b), "partially resolved", was selected, while the written response states that the concerns have been addressed and does not include follow-up questions. We wanted to mention this in case the selection was unintended.
> > >
> > > We believe that the additional experiments on other 1D ordered tokenizers (Semanticist) and scale-wise models (Infinity), along with the theoretical analysis and clarifications, substantially address the concerns raised in the original review. We will incorporate these into the revised paper and release the models and code used in this work to support reproducibility and further research. We sincerely thank the reviewer again for the constructive and thoughtful feedback.

---

### Official Review · Reviewer_N3iF · 2026-03-12

**Soundness:** 3
**Presentation:** 3
**Significance:** 2
**Originality:** 2
**Overall Recommendation:** 4
**Confidence:** 5

**Summary:**

The paper studies the problem of test-time search for image generation models. Particularly, the paper compares the test-time search between image generation models with 1D tokens and 2D tokens. By using various test-time search methods like beam-search and using various verifiers, the paper shows that the 1D-token-image-generation model outperforms the 2D ones. The paper also demonstrates some interesting use cases of the test-time search such as training-free text-to-image generation with guidance.

**Compliance With Llm Reviewing Policy:**

Affirmed.

**Final Justification:**

The rebuttal has addressed the main concerns.

**Key Questions For Authors:**

Please address the weaknesses.

**Limitations:**

yes

**Strengths And Weaknesses:**

Strengths:
1. The paper studies and compares 1D and 2D-token image generation models from a new perspective: to test the models' behavior in test-time search tasks.
2. The paper also shows some interesting use cases of test-time search.

Weaknesses:
1. While being useful as reported, the test-time search takes significantly more time/computation compared to baseline generation. The paper only studies the effects of test-time search and is more like a benchmark. However, to make this method really useful in practice, I think more efforts should be put into reducing the test-time search computation, such that with moderate overhead, we can make the 1D token image generation model much more useful in practice.

2. The search method for the 2D token case could potentially be improved. Currently, all the remaining tokens will be set as padding tokens. Maybe a more reasonable approach is to use some lighter-weight generation models to do in-filling for the remaining tokens (e.g., low-resolution parts of a VAR model) and search based on that. Because of this, I think the paper's current finding is also not surprising at all: we add very little information in every step for the 2D generation model, and therefore, the search is not effective.

3. The sacle of the main experiment is limited: only 300 images on COCO dataset.

---

> ### Author Rebuttal · Authors · 2026-03-31
>
> We sincerely thank the reviewer for the constructive feedback and for recognizing the novelty of our perspective. We address your concerns below.
>
> # Compute overhead and practical utility (W1)
>
> **Search provides flexible capability-compute tradeoffs.** The goal of test-time search is not to replace baseline generation but to offer a flexible tradeoff: practitioners allocate more compute when higher quality is needed, and less otherwise. Even a small search budget yields noticeable gains. (cf. Figure 6).
>
> **Search can substitute for training-time compute.** A key practical benefit is that test-time search can compensate for reduced training compute, a well-established phenomenon in language modeling (Snell et al., 2024) and game-playing AI (Silver et al. 2017). As shown in Figure 8, a smaller 530M-parameter model with sufficient test-time search matches the performance of a 3.4B-parameter model without search at the same inference FLOPs, a meaningful tradeoff when training is expensive but inference compute is flexible.
>
> **Our work also contributes to efficiency.** By establishing that 1D ordered tokens are fundamentally more amenable to search, we provide the representational foundation that future efficient search methods can build on. As a concrete step, we explored an adaptive early-stopping variant that determines online how many tokens to search over based on verifier score saturation, reducing average generation time by ~35% and average token usage by ~67% (from 256 to ~84 tokens), with accuracy remaining within 0.5%.
>
> # Fair comparison and the 2D infilling baseline (W2)
>
> We thank the reviewer for this suggestion. It is closely related to lookahead search, which we have already studied: lookahead rolls out remaining tokens using the AR prior before applying the verifier, achieving a similar effect to infilling.
>
> As shown in Table B1, **lookahead search partially compensates for beam search on 2D grid tokens, but at substantially higher computational cost** (~200K NFE vs. ~2K NFE for beam search at 32 steps). Using a lightweight infilling model could reduce this cost, but would require training a distilled model on the same token space, introducing additional training compute and sacrificing the training-free nature of our approach. In contrast, FlexTok with more beam search steps outperforms 2D grid tokens with lookahead using far fewer NFEs.
>
> **Critically, even with lookahead, the gap between 1D and 2D tokens is not fully closed.** For training-free generation via pure search, lookahead cannot compensate, as 2D grid tokens lack the semantic intermediate states required for effective ordering. For zero-shot multimodal control on DreamBench++, FlexTok with beam search achieves +18% on DINO-I, while Janus with lookahead achieves only +6%, despite both using 32 search steps. This demonstrates that the advantage of 1D ordered tokens reflects a fundamental representational property that cannot be mitigated by the choice of search algorithm alone. We also evaluate Infinity (a VAR-style model with scale-wise ordering), which achieves +6.2% CLIPScore improvement, larger than Janus (+5.3%) but less than FlexTok (+9.6%), further confirming that different ordering schemes have meaningfully different impacts on search effectiveness. Full results are in Table A2 in response to Reviewer FHzW.
>
> **Table B1.** Beam search vs. lookahead search at the same number of steps (CLIPScore % on COCO).
>
> |  | FlexTok | 2D grid tok. | NFE |
> | --- | --- | --- | --- |
> | AR | 80.39 | 79.06 | 256 |
> | Beam Search (step=32) | 93.44 | 81.59 | 2,880 |
> | Lookahead Search (step=32) | 93.65 | 96.14 | 201,280 |
> | Beam Search (step=64) | **98.33** | 81.87 | 4,480 |
>
> # Experimental scale (W3)
>
> We limited our controlled ablations to 300 COCO images due to the substantial computational cost of comprehensive hyperparameter sweeps and lookahead baselines. To address the concern about statistical significance, we scaled our key beam search evaluations to a **1,000-image COCO subset**. As shown in Table B2, variance across 5 random subsets is low, and the performance gap between 1D and 2D tokens remains fully consistent with our 300-image findings. We additionally evaluate on GenEval (553 captions), where the same conclusion holds across both ImageReward and GenEval accuracy metrics. Results are shown [here](https://anonymous.4open.science/r/sot-BF72/tts_geneval.png).
>
> **Table B2.** Variance analysis with CLIPScore (%) on COCO (300 vs. 1K images). Δ and its standard deviation are computed from per-run differences.
>
> | Algorithm| 300 Images (Score ↑) | 1K Images (Score ↑) | 300 Images (Mean ± Std, 5 runs) |
> | --- | --- | --- | --- |
> | FlexTok Baseline | 80.39 | 80.28 | 80.23 ± 0.38 |
> | FlexTok Beam Search | 93.44 (Δ 13.05) | 93.72 (Δ 13.44) | 93.53 ± 0.50 (Δ 13.30 ± 0.29) |
> | 2D Grid Baseline | 79.06 | 79.19 | 79.25 ± 0.24 |
> | 2D Grid Beam Search | 81.59 (Δ 2.53) | 81.59 (Δ 2.40) | 81.68 ± 0.35 (Δ 2.43 ± 0.24) |

---

> > ### Author Rebuttal · Reviewer_N3iF · 2026-04-02
> >
> > My concerns are addressed. I raise the score to 4.

---

> > > ### Author Response · Authors · 2026-04-08
> > >
> > > We sincerely thank the reviewer for the thoughtful feedback and the positive acknowledgement. We are glad that the clarifications helped address the concerns. We will incorporate discussion of compute overhead, lookahead search, and experimental scale into the final revision and release the models and code to support reproducibility and further research.

---

### Official Review · Reviewer_AhiD · 2026-03-13

**Soundness:** 2
**Presentation:** 3
**Significance:** 2
**Originality:** 2
**Overall Recommendation:** 3
**Confidence:** 4

**Summary:**

This paper investigates why test-time search has historically struggled in autoregressive image generation. The authors hypothesize that the token ordering structure is the key bottleneck. Through experiments with a highly compressed 1D ordered tokenizer (FlexTok), they demonstrate that such structures enable effective test-time search, significantly improving generation quality without additional training. They show that search alone on these 1D tokens can outperform standard autoregressive decoding, suggesting that the token structure itself encodes sufficient semantic priors for generation.

**Compliance With Llm Reviewing Policy:**

Affirmed.

**Final Justification:**

Thanks for the rebuttal. I think the concerns have been resolved.

**Key Questions For Authors:**

It is suggested to analyze the limitations of this method through some failure cases.

**Limitations:**

Yes.

**Strengths And Weaknesses:**

**Strength**
1. The paper provides compelling empirical evidence that 1D token significantly impacts the effectiveness of test-time search. By demonstrating that a switch to 1D ordered tokens enables beam search to drastically improve generation quality.
2. A key strength is the practicality of the proposed approach. Achieving impressive results purely through test-time search on existing models—without any additional training or architectural changes.

**Weakness**
1. The core claim that '1D ordered tokens suit test-time search better' feels quite intuitive given their coarse-to-fine nature. Without a new algorithm or theoretical analysis, it’s unclear if this empirical validation offers a novel insight or just confirms common sense. Could the authors clarify why this finding isn't trivial or perhaps provide some theoretical intuition (e.g., information density) to deepen the contribution?
2. The conclusion relies heavily on one model(FlexTok). Since the base 2D model already underperforms the 1D one, it's hard to tell if the search gain comes from the token structure or simply because the 1D tokenizer is inherently better. Would this advantage hold with stronger 2D baselines or different 1D variants? Broader comparisons are helpful to rule out model-specific artifacts.

---

> ### Author Rebuttal · Authors · 2026-03-31
>
> We sincerely thank the reviewer for the thoughtful and constructive review. We appreciate that the reviewer found our empirical evidence compelling and our approach practical. We address the three main concerns below.
>
> # Contribution and theoretical intuition (W1)
>
> **Why this is not trivial:** While the hypothesis that coarse-to-fine token structures may benefit search is intuitive, its implications were previously unclear and inconcrete. Our work provides the first systematic study of how token structure affects test-time search in autoregressive image generation. Prior to our work, it was unclear: (1) how large the benefit of ordered tokens is and how it depends on the choice of search algorithms; (2) whether search alone, without any AR prior, can produce coherent images under ordered tokenization using standard search algorithms; and (3) whether inference-time search can enable new capabilities such as zero-shot multimodal control when the model is trained only for text-to-image generation.
>
> Our experiments demonstrate that all three effects occur consistently under ordered tokenization, and offer a new perspective on tokenization design: token structure can fundamentally affect test-time scaling behavior, beyond reconstruction or generation quality.
>
> **Theoretical intuition.** The efficacy of guided search depends on heuristic error [b1], which measures how well a partial sequence predicts the final verifier score. This is closely related to prefix information density: for search to be effective, early prefixes must be informative about the final sequence. 1D ordered tokenizers achieve this through nested dropout, encouraging early tokens to capture global semantics (analogous to PCA [b2], where early components provide strong signal). In contrast, 2D grid tokenizations distribute information uniformly, so early partial sequences carry little predictive signal, making pruning ineffective. We will incorporate this discussion into the revision. A more formal analysis is provided in our response to Reviewer BUbr.
>
> # Fair comparison and other 1D variants (W2)
>
> **Clarification for the controlled study.** In our controlled study, the 1D FlexTok and 2D grid token models show similar base performance without search (CLIPScore 0.80 vs. 0.79), with matched data, architecture, and compute. The gap emerges only after applying test-time search, especially for beam search (cf. Figure 6), confirming the advantage comes from token structure rather than base model strength.
>
> **Stronger 2D baseline.** We compare against Janus, a competitive 2D grid model that actually *outperforms* FlexTok slightly when generation without search (CLIPScore ~0.82 vs. ~0.80). However, as inference compute scales, FlexTok's ordered structure benefits from beam search far more effectively, eventually reaching ~1.11 vs. Janus plateauing at ~0.90—a margin of 0.21. This confirms the advantage holds even against stronger 2D baselines.
>
> **Additional 1D variant.** We conduct experiments using **Semanticist** (Wen et al., 2025), which enforces ordering through a similar nested dropout mechanism but differs in architecture and token space design. Although Semanticist has slightly lower base AR performance, it scales significantly better than LlamaGen (a comparable 2D tokenizer) under beam search: on simple prompts, beam search improves Semanticist by +10.42% vs. +3.51% for LlamaGen; on complex prompts, +12.45% vs. +4.04%. Full results are in our response to reviewer FHzW (Table A1).
>
> # Failure cases (Question)
>
> We have examined representative failure cases and identified two primary failure modes, to be included with visual examples in a dedicated appendix section.
>
> **Verifier hacking.** When the search budget is large, optimization may overfit to the verifier and exploit its blind spots, producing visually implausible images that nonetheless achieve high scores. For example (Fig. 13), optimizing CLIP or Grounded-SAM scores can degrade aesthetic quality, while optimizing aesthetic score alone can reduce GenEval accuracy. In Fig. 14, for the prompt *“a vase above a fire hydrant”*, optimizing for aesthetics produces visually appealing images but weakens the presence of the vase. Verifier ensembles partially alleviate this issue, but improving verifier robustness remains an important direction for future work.
>
> **Prior bottleneck.** Test-time search cannot recover information missing from the AR prior. In Figure 10, under a uniform prior, search fails to generate key semantic elements (e.g., "wine") due to weak object priors. In Figure 9, searched results still deviate from the reference image (e.g., "fog" case)—search improves over AR decoding but remains limited by a misaligned prior.
>
> [b1] Pearl, J. *Heuristics: Intelligent Search Strategies for Computer Problem Solving.* Addison-Wesley, 1984.
> [b2] Rippel, Oren et al. "Learning ordered representations with nested dropout." *International Conference on Machine Learning*. PMLR, 2014.

---

> > ### Author Rebuttal · Reviewer_AhiD · 2026-04-04
> >
> > Thanks for your rebuttal. My concerns have been partially resolved. I will adjust my score considering with the novelty and contribution.

---

> > > ### Author Response · Authors · 2026-04-08
> > >
> > > We thank the reviewer for the acknowledgement and for noting that the concerns have been partially resolved. We appreciate the reviewer's reconsideration of the paper's novelty and contribution. Since the remaining concerns were not specified in detail, we briefly address the two main issues raised in the original review and summarize how our rebuttal resolves them.
> > >
> > > > The core claim that '1D ordered tokens suit test-time search better' feels quite intuitive given their coarse-to-fine nature. ... Could the authors clarify why this finding isn't trivial or perhaps provide some theoretical intuition (e.g., information density) to deepen the contribution?
> > > >
> > >
> > > We agree that the hypothesis may appear intuitive in hindsight, but we believe the paper makes several non-trivial contributions beyond confirming intuition:
> > >
> > > 1. **Systematic study of token structure and search:**  To our knowledge, this is the first work to systematically study how different search algorithms interact with different token structures, and to quantify the magnitude of this effect under controlled settings. Our experiments show that the advantage of 1D ordered tokens is substantial and consistent across different AR models and cannot be closed by the choice of search algorithm alone (cf. Figure 6), suggesting that token structure is a fundamental factor rather than an artifact of a specific setup.
> > > 2. **Training-free generation via standard beam search:** We show that standard discrete search methods (e.g., beam search), when applied to ordered token sequences, enable meaningful training-free generation without an autoregressive model, a capability that does not emerge with traditional 2D grid tokens. This represents a qualitatively new capability that is not implied by the intuition alone.
> > > 3. **Verifier-guided multimodal control**: We show that search enables flexible zero-shot multimodal control using reference images and image-based verifiers, and that this capability is more effective with ordered tokens than with grid-based tokenizations.
> > > 4. **Theoretical grounding.** In our rebuttal to Reviewer BUbr, we provided a formal analysis connecting the search gap to intermediate reconstruction error. Ordered tokenizers trained with nested dropout maintain low reconstruction error throughout generation, leading to smaller heuristic error and more reliable early-stage pruning. This provides a principled explanation beyond intuition and will be included in the revision.
> > >
> > > > The conclusion relies heavily on one model(FlexTok). Since the base 2D model already underperforms the 1D one, it's hard to tell if the search gain comes from the token structure or simply because the 1D tokenizer is inherently better. Would this advantage hold with stronger 2D baselines or different 1D variants? Broader comparisons are helpful to rule out model-specific artifacts.
> > > >
> > >
> > > We would like to clarify that the premise that the base 2D model underperforms the 1D model does not hold in our experiments. In our controlled setting, the 1D and 2D models have **comparable base performance without search** (CLIPScore 0.80 vs. 0.79, with matched data, architecture, and compute). The large gap emerges only after applying test-time search (cf. Figure 6), confirming that the advantage comes from the searchability of the token structure rather than stronger base generation quality. Furthermore, **Janus**, a competitive 2D baseline, actually **outperforms** FlexTok slightly without search (~0.82 vs. ~0.80), yet FlexTok benefits far more from beam search and ultimately surpasses Janus under increased inference compute (cf. Figure 7).
> > >
> > > To further rule out model-specific artifacts, we conducted additional experiments during the rebuttal using **Semanticist**, another 1D ordered token-based model, and compared it against LlamaGen, a comparable 2D grid token-based model. The same trend holds: beam search improves Semanticist by +10.42% vs. +3.51% for LlamaGen on simple prompts, and +12.45% vs. +4.04% on complex prompts (Table A1 in our response to Reviewer FHzW). These three lines of evidence (controlled comparison, stronger 2D baseline, and a different 1D tokenizer) consistently show that the advantage is structural rather than model-specific.
> > >
> > > We believe the additional experiments, theoretical analysis, and clarifications substantially address the concerns raised in the original review. We will incorporate the changes to the paper and release the models and code used in this work to support reproducibility and further research. We also note that the reviewer mentioned in the acknowledgement that the score would be adjusted based on novelty and contribution, and we would be grateful if the reviewer could confirm whether the current evidence warrants an update.

---

### Official Review · Reviewer_FHzW · 2026-03-13

**Soundness:** 3
**Presentation:** 3
**Significance:** 3
**Originality:** 3
**Overall Recommendation:** 4
**Confidence:** 4

**Summary:**

The paper analyzes 1D vs. 2D tokenization, and shows that 1D ordered tokens enable more effective test-time search in autoregressive image generation through a coarse-to-fine token hierarchy, even allowing reasonable generation with search alone without an AR transformer.

**Compliance With Llm Reviewing Policy:**

Affirmed.

**Ethical Review Concerns:**

No need to ethical review.

**Final Justification:**

The paper is well written and presents an interesting perspective on test-time scaling from the viewpoint of token structure. The observations are intriguing and supported by thorough experiments, and the analysis may inspire further research in this direction.

In the rebuttal, the authors provided additional experiments and clarifications addressing my concerns on generality and the relationship to prior work. These responses are convincing and strengthen the overall contribution of the paper.

Overall, I find the work insightful and well-executed, and I am satisfied that the main concerns have been adequately addressed.

**Key Questions For Authors:**

See weaknesses.

**Limitations:**

No. The paper would benefit from a brief discussion of potential limitations.

**Strengths And Weaknesses:**

**Strengths:**
- The paper is clearly written and well motivated. It studies the impact of token structure on test-time scaling (TTS), providing an interesting perspective from the tokenization level rather than purely algorithmic improvements.
- The observations presented in the paper are intriguing. For example, the finding that reasonable generation can be achieved with search alone under an ordered 1D tokenization scheme, as well as the comparison of scalability with respect to NFE across different TTS strategies, may inspire further research in this direction.
- Experiments and analyses are thorough and well organized, providing detailed evidence to support the paper’s observations.

**Weaknesses:**
- Most of the observations and experiments appear to be conducted on the FlexTok tokenizer, which is trained with token dropping to enforce a coarse-to-fine semantic ordering. If the method relies on such tokenizers with explicit hierarchical structure, its applicability to more general tokenization schemes may be limited.
- The experiments are mainly conducted on the FlexTok tokenizer, which enforces a coarse-to-fine token hierarchy during training. It would strengthen the paper to examine whether similar observations hold for other generation paradigms, such as 2D tokenizers with mask-based generation (e.g., Show-o[1]) or scale-wise autoregressive frameworks like VAR[2].
- Related work has explored the editability and manipulability of 1D token spaces [3], which may partially overlap with the observations in this paper. It would be helpful to clarify the relationship between the two.

**References:**

[1] Show-o: One Single Transformer to Unify Multimodal Understanding and Generation

[2] Visual Autoregressive Modeling: Scalable Image Generation via Next-Scale Prediction

[3] Highly Compressed Tokenizer Can Generate Without Training

---

> ### Author Rebuttal · Authors · 2026-03-31
>
> We sincerely thank the reviewer for the thoughtful and encouraging feedback. We are glad you found the paper clearly written and the observations on token structure and test-time scaling interesting. We agree the scope of our evidence should be clarified: our claim is not that search only works for FlexTok or that all tokenizations behave similarly, but that token structure materially affects searchability, and that ordered coarse-to-fine tokenizations are especially amenable to search.
>
> # Broader tokenization scheme. (W1&W2)
> We agree this is an important question. Our intended claim is comparative rather than exclusive: search can improve multiple generation schemes, but to very different extents depending on whether intermediate prefixes are semantically meaningful enough for verifiers to guide generation. This is already the picture suggested by the current submission: compared with controlled 2D grid tokenization and Janus, search helps both, but beam search is much more effective for the 1D ordered structure.
>
> To test generality beyond FlexTok, we evaluate Semanticist (Wen et al., 2025), another 1D ordered tokenizer, against LlamaGen-L with a 2D grid tokenizer on class-to-image generation on ImageNet-1K (Table A1). The same trend holds: all search methods help both models, but *beam search benefits the ordered 1D tokenizer much more* (e.g., +10.4 vs. +3.5 CLIPScore on simple prompts; +12.5 vs. +4.0 on complex prompts), supporting that the advantage comes from token structure rather than a specific implementation. Example visualizations are provided [here](https://anonymous.4open.science/r/sot-BF72/semanticist_vis.pdf).
>
> - Experimental setting: Since Semanticist is trained for ImageNet class-to-image generation, we use two prompt types for CLIPScore: (1) simple prompts (“a photo of a [CLASS_NAME]”) and (2) enriched captions from ImageNet-1K-VL-Enriched [a1] to test scaling with richer guidance and probe text-to-image under weak class priors. We apply identical settings to both models: Best-of-N (N=10), beam search (beam=5, candidates=10, 4 steps), and lookahead search (same as beam with full rollout).
>
> Table A1: Semanticist (1D) vs. LlamaGen (2D) on ImageNet-1K (CLIPScore %, 1K images).
> | Model | Tokenizer Type | Prompt Type | Base | Best-of-N | Beam Search | Lookahead Search |
> | --- | --- | --- | --- | --- | --- | --- |
> | Semanticist  | 1D ordered | Simple | 74.60 | 81.77 (+7.16) | **85.02 (+10.42)** | 85.51 (+10.91) |
> | LlamaGen | 2D grid | Simple | 76.85 | 82.50 (+5.64) | 80.36 (+3.51) | 81.96 (+5.11) |
> | Semanticist | 1D ordered | Complex | 70.67 | 79.72 (+9.06) | **83.12 (+12.45)** | 83.68 (+13.02) |
> | LlamaGen | 2D grid | Complex | 73.52 | 80.48 (+6.96) | 77.56 (+4.04) | 79.42 (+5.90) |
>
> We also evaluated **Infinity** [a2], a scale-wise autoregressive paradigm related to VAR (Table A2). Under matched search budgets (beam=5, candidates=10, 9 steps), Infinity also benefits substantially from beam search (+6.2 CLIPScore), more than Janus (+5.3) but less than FlexTok (+9.6). We think this is a useful intermediate result: *having meaningful ordering helps search, but semantic coarse-to-fine ordering helps the most.*
>
> Table A2. Comparison of generation paradigms on COCO (CLIPScore, %).
> | Model | Janus | Infinity | FlexTok |
> | --- | --- | --- | --- |
> | Token Structure | 2D grid | 2D multi-scale | 1D ordered |
> | AR | 82.3 | 81.9 | 80.4 |
> | Beam Search | 87.6 (+5.3) | **88.1 (+6.2)** | 90.0 (+9.6) |
>
> We will add these results and revise the paper accordingly, leaving broader validation across more paradigms as future work.
>
> # Related work (W3)
> Thank you for pointing out this highly relevant work. We will add a clearer discussion in the revision. We view the connection as close in spirit but different in mechanism and scope. [3] study training-free generation/editing via continuous optimization in highly compressed 1D token spaces, whereas our paper studies discrete sequential search over token sequences, with a focus on test-time scaling, verifier-guided generation, and the interaction with autoregressive priors. While there is conceptual overlap at the level of token-space generation and in demonstrating training-free generation, the central questions and methodologies differ.
>
> # Limitations
> We agree that the paper would benefit from a clearer discussion of limitations. We will add: (1) current empirical focus on a limited set of tokenizers/paradigms, (2) dependence on verifier robustness and possible verifier hacking, and (3) search efficiency, since this paper uses classical search mainly as an analysis tool rather than proposing compute-optimal search methods.
>
> [a1] Visual Layer. *ImageNet-1K-VL-Enriched Dataset*. https://huggingface.co/datasets/visual-layer/imagenet-1k-vl-enriched.
> [a2] Han, Jian, et al. "Infinity: Scaling bitwise autoregressive modeling for high-resolution image synthesis." *CVPR*. 2025.

---

> > ### Author Rebuttal · Reviewer_FHzW · 2026-04-03
> >
> > Thank you for the detailed rebuttal and additional experiments, which clearly addressed my concerns and improved my understanding of the paper. Overall, I find this to be a well-executed work, and I support its acceptance at ICML.

---

> > > ### Author Response · Authors · 2026-04-08
> > >
> > > We sincerely thank the reviewer for the thoughtful feedback and the positive acknowledgement. We are glad that the additional experiments and clarifications helped address the concerns. We will incorporate the broader comparisons, discussion of related work, and limitations into the final revision and release the models and code to support reproducibility and further research.

---

### Decision · Program_Chairs · 2026-04-30

**Decision:**

Accept (regular)

**Comment:**

This paper studies how tokenization structure affects the effectiveness of test-time search in autoregressive image generation. It shows that 1D coarse-to-fine ordered tokens enable significantly better test-time scaling than conventional 2D grid tokens, due to more informative intermediate states for verifier-guided search.
Reviewers agree that the paper is well-written and provides strong, systematic empirical evidence. The main concerns focus on limited conceptual novelty, reliance on specific tokenizers, and lack of strong theoretical grounding.
The rebuttal addresses these concerns effectively by adding results on additional tokenizers and stronger baselines, and by providing clearer intuition for why ordered tokens improve search. Several reviewers indicate their concerns are fully or largely resolved.
Overall, while the contribution is primarily empirical and the theoretical analysis remains preliminary, the paper raises a meaningful and underexplored question about the interaction between representation and test-time scaling, and supports it with convincing evidence.